# Supervised Knowledge May Hurt Novel Class Discovery Performance

**Ziyun Li**                                                                    *ziyun.li@hpi.de*
*Hasso Plattner Institute*
*University of Potsdam*

**Jona Otholt**                                                                 *jona.Otholt@hpi.de*
*Hasso Plattner Institute*
*University of Potsdam*

**Ben Dai**[*]                                                                  *bendai@cuhk.edu.hk*
*Department of Statistics*
*Chinese University of Hong Kong*

**Di Hu**                                                                       *dihu@ruc.edu.cn*
*Gaoling School of Artificial Intelligence*
*Renmin University of China*

**Christoph Meinel**                                                            *christoph.meinel@hpi.de*
*Hasso Plattner Institute*
*University of Potsdam*

**Haojin Yang**[*]                                                              *haojin.yang@hpi.de*
*Hasso Plattner Institute*
*University of Potsdam*

**Reviewed on OpenReview:** *https://openreview.net/forum?id=oqOBTo5uWD*

## Abstract

Novel class discovery (NCD) aims to infer novel categories in an unlabeled dataset by leveraging prior knowledge of a labeled set comprising disjoint but related classes. Given that most existing literature focuses primarily on utilizing supervised knowledge from a labeled set at the methodology level, this paper considers the question: *Is supervised knowledge always helpful at different levels of semantic relevance?* To proceed, we first establish a novel metric, so-called *transfer flow*, to measure the semantic similarity between labeled/unlabeled datasets. To show the validity of the proposed metric, we build up a large-scale benchmark with various degrees of semantic similarities between labeled/unlabeled datasets on ImageNet by leveraging its hierarchical class structure. The results based on the proposed benchmark show that the proposed *transfer flow* is in line with the hierarchical class structure; and that NCD performance is consistent with the semantic similarities (measured by the proposed metric). Next, by using the proposed *transfer flow*, we conduct various empirical experiments with different levels of semantic similarity, yielding that *supervised knowledge may hurt NCD performance*. Specifically, using supervised information from a low-similarity labeled set may lead to a suboptimal result as compared to using pure self-supervised knowledge. These results reveal the inadequacy of the existing NCD literature which usually assumes that supervised knowledge is beneficial. Finally, we develop a pseudo-version of the *transfer flow* as a practical reference to decide if supervised knowledge should be used in NCD. Its

---

[*]Corresponding author

effectiveness is supported by our empirical studies, which show that the pseudo *transfer flow* (with or without supervised knowledge) is consistent with the corresponding accuracy based on various datasets. Code is released at `https://github.com/J-L-O/SK-Hurt-NCD`

## 1 Introduction

The combination of data, algorithms, and computing power has resulted in a boom in the field of artificial intelligence, particularly supervised learning with its large number of powerful deep models. These deep models are capable of properly identifying and clustering classes that are present in the training set (i.e., known/seen classes), matching or surpassing human performance. However, they lack reliable extrapolation capacity when confronted with novel classes (i.e., unseen classes) while humans can easily recognize the novel categories. A classic illustration is how effortlessly a person can readily discriminate (cluster) unseen but similar vehicles (e.g., trains and cars) based on prior experience. This motivated researchers to develop a challenge termed novel class discovery (NCD) (Han et al., 2019; Chi et al., 2022; Han et al., 2021; Zhong et al., 2021a), with the goal of discovering novel classes in an unlabeled dataset by leveraging knowledge from a labeled set, which contains related but disjoint classes.

Recently, the majority of works on NCD implicitly assumes that more data is better, and devotes to designing and developing neural networks to better utilize the supervised knowledge contained in the labeled set. For example, DTC Han et al. (2019), RS Han et al. (2021; 2020) and RSMKD Zhao & Han (2021) transfer supervised knowledge by pretraining the model on the labeled data. OpenMix Zhong et al. (2021b)mixes supervised knowledge from the labeled set and unsupervised knowledge from the unlabeled set to learn a joint label distribution. UNO Fini et al. (2021) combines supervised knowledge in a unified objective function.

Despite their remarkable performance, there is a less in-depth analysis of the supervised knowledge from the labeled set itself. Therefore, in this work, we investigate a fundamental question of NCD: *Is supervised knowledge always helpful?* Clearly, this question is naturally associated with the discrepancy between labeled and unlabeled sets. As mentioned by Chi et al. (2022), NCD is theoretically solvable when labeled and unlabeled sets share high-level semantic features. However, no quantitative analysis of semantic similarity was proposed. In this paper, we hypothesize that supervised knowledge would be beneficial when labeled and unlabeled sets share a high degree of semantic similarity but may be less beneficial *or even harmful* when semantic similarity is low. To examine the hypothesis, we first propose a quantitative metric, *transfer flow*, to measure the semantic label similarity of the two datasets. Specifically, it quantifies the discriminative information in the labeled set leaked in the unlabeled set, i.e., how much information we can leverage from the labeled dataset to help improve its performance. More details are provided in Section 3.

To demonstrate the validity of *transfer flow*, we establish a new *NCD benchmark* with multiple semantic similarity levels. Specifically, the new benchmark is constructed based on a large-scale dataset, ImageNet (Deng et al., 2009), by leveraging its hierarchical semantic information. It includes three difficulty levels, high, medium, and low semantic similarity, where each difficulty level includes two data settings. Based on the benchmark, our experiments confirm that semantic similarity is positively related with NCD performance on multiple pairs of labeled and unlabeled sets with varying semantic similarity under multiple baselines (Han et al., 2019; 2021; Fini et al., 2021; MacQueen et al., 1967). Also, a mutual validation between the proposed metric and the benchmark is conducted, which reveals that the *transfer flow* corresponds strongly with NCD performance. Detailed information on the benchmark can be found in Section 4.

Based on the proposed metric and benchmark, we then analyze our core research question. Our experiments are conducted on the current state-of-the-art (SOTA) approaches (Fini et al., 2021; Zhong et al., 2021a; Han et al., 2021), comparing the NCD performance with (i.e., standard NCD) or without supervised information. Unexpectedly but reasonably, we observe that the latter can outperform the former in cases where the semantic similarity between the labeled and unlabeled sets is low, in contrast to the commonly held assumption that supervised knowledge (or more data) can improve NCD performance. As a by-product, this also raises a question of whether to use supervised or just self-supervised knowledge for a given dataset. To address this issue, we provide two practical solutions. (i) A data selection solution. We develop a pseudo-version of the proposed metric, namely *pseudo transfer flow*, as a practical metric to infer the flow of supervised knowledge

and/or self-supervised knowledge. Thus, it serves as an instructive reference to decide what sort of data we intend to employ. (ii) A data combining solution. We develop a new model which smoothly combines supervised and self-supervised knowledge from the labeled set and achieves 3% and 5% improvement in both CIFAR100 (Krizhevsky, 2009) and ImageNet compared to SOTA. More information can be found in Section 5.

We summarize our contributions as follows:

- We find that using supervised knowledge from the labeled set may lead to suboptimal performance in low semantic NCD datasets. Based on this finding, we propose two practical methods and achieve ∼3% and ∼5% improvement in both CIFAR100 and ImageNet compared to SOTA.

- We introduce a theoretically reliable metric to measure the semantic similarity between labeled and unlabeled sets. A mutual validation is conducted between the proposed metric and a benchmark, which suggests that the proposed metric strongly agrees with NCD performance.

- We establish a comprehensive benchmark with varying degrees of difficulty based on ImageNet by leveraging its hierarchical semantic similarity. Besides, we empirically confirm that semantic similarity is indeed a significant factor influencing NCD performance.

## 2 Related Work

Novel class discovery (NCD) is a relatively new problem proposed in recent years, aiming to discover novel classes (i.e., assign them to several clusters) by making use of similar but different known classes. Unlike unsupervised learning, NCD also requires labeled known-class data to help cluster novel-class data. NCD is first formalized in DTC (Han et al., 2019), but the study of NCD can be dated back to earlier works, such as KCL (Hsu et al., 2018) and MCL (Hsu et al., 2019). Both of these methods are designed for general task transfer learning and connect two models trained with labeled data and unlabeled data, respectively. In contrast, DTC first learns a data embedding on the labeled data with metric learning, then employs a deep embedded clustering method based on Xie et al. (2016) to cluster the novel-class data.

More recent works, such as RS (Han et al., 2021; 2020) and RSMKD (Zhao & Han, 2021), use self-supervised learning to boost feature extraction and use the learned features to obtain pairwise similarity estimates. Additionally, Zhao & Han (2021) improve RS by using information from both local and global views, as well as mutual knowledge distillation to promote information exchange and agreement. NCL (Zhong et al., 2021a) extracts and aggregates the pairwise pseudo-labels for the unlabeled data via contrastive learning and generates hard negatives by mixing the labeled and unlabeled data in a feature space. This idea of mixing data is also used in OpenMix (Zhong et al., 2021b), which mixes known-class and novel-class data to learn a joint label distribution. The current state-of-the-art, UNO (Fini et al., 2021), combines pseudo-labels with ground-truth labels in a unified objective function that enables better use of synergies between labeled and unlabeled data without requiring self-supervised pretraining. The most related one to our work is Meta discovery (Chi et al., 2022), which demonstrated the solvability of NCD by showing high-level semantic similarities between known and unknown classes. However, they lacked quantitative analysis, which we addressed by introducing a metric for quantifying semantic similarity. In addition, Chi et al. (2022) concentrate on developing a method for scenarios with limited novel class data, while our study focuses on the standard NCD setting.

## 3 Quantifying Semantic Similarity

In this section, we present a novel metric for measuring the semantic similarity between labeled and unlabeled sets.

### 3.1 NCD Framework

We denote $(\mathbf{X}_l, Y_l)$ and $(\mathbf{X}_u, Y_u)$ as random samples under the *labeled/unlabeled probability measures* $\mathbb{P}_{\mathbf{X},Y}$ and $\mathbb{Q}_{\mathbf{X},Y}$, respectively. $\mathbf{X}_l \in\subset \mathbb{R}^d$ and $\mathbf{X}_u \in \mathcal{X}_u \subset \mathbb{R}^d$ are the labeled/unlabeled feature vectors, $Y_l \in \mathcal{C}_l$ and

$Y_u \in \mathcal{C}_u$ are the true labels of labeled/unlabeled data, where $\mathcal{C}_l$ and $\mathcal{C}_u$ are the label sets under the labeled and unlabeled probability measures $\mathbb{P}_{\mathbf{X},Y}$ and $\mathbb{Q}_{\mathbf{X},Y}$, respectively. Given a labeled set $\mathcal{L}_n = (\mathbf{X}_{l,i}, Y_{l,i})_{i=1}^n$ independently drawn from the labeled probability measure $\mathbb{P}_{\mathbf{X},Y}$, and an unlabeled dataset $\mathcal{U}_m = (\mathbf{X}_{u,i})_{i=1}^m$ independently drawn from the unlabeled probability measure $\mathbb{Q}_{\mathbf{X}_u}$, our primary goal is to predict $Y_{u,i}$ given $\mathbf{X}_{u,i}$, where $Y_{u,i}$ is the label of the $i$-th unlabeled sample $\mathbf{X}_{u,i}$. We now give a general definition of NCD.

**Definition 1 (Novel Class Discovery)** *Let $\mathbb{P}_{\mathbf{X}_l,Y_l}$ be a labeled probability measure on $\mathcal{X}_l \times \mathcal{C}_l$, and $\mathbb{Q}_{\mathbf{X}_u,Y_u}$ be an unlabeled probability measure on $\mathcal{X}_u \times \mathcal{C}_u$, with $\mathcal{C}_u \cap \mathcal{C}_l = \emptyset$. Given a labeled dataset $\mathcal{L}_n$ sampled from $\mathbb{P}_{\mathbf{X}_l,Y_l}$ and an unlabeled dataset $\mathcal{U}_m$ sampled from $\mathbb{Q}_{\mathbf{X}_u}$, novel class discovery aims to predict the label $Y_u$ of each unlabeled instance $X_u$ given $\mathcal{L}_n$ and $\mathcal{U}_m$.*

### 3.2 Transfer Flow

For further investigating the effectiveness of supervised knowledge in NCD, we propose a quantitative metric, namely *transfer flow*, to assess semantic similarity between labeled/unlabeled datasets. To the best of our knowledge, the question of how to measure the semantic similarity between the labeled and unlabeled sets in NCD remains unsolved.

To proceed, we begin with introducing Maximum Mean Discrepancy (MMD; Gretton et al. (2012)), which is used to measure the discrepancy of two distributions. For example, MMD of two random variables $\mathbf{Z} \sim \mathbb{P}_{\mathbf{Z}}$ and $\mathbf{Z}' \sim \mathbb{P}_{\mathbf{Z}'}$ is defined as:

$$\mathrm{MMD}_{\mathcal{H}}\big(\mathbb{P}_{\mathbf{Z}}, \mathbb{P}_{\mathbf{Z}'}\big) := \sup_{\|h\|_{\mathcal{H}} \leq 1} \Big( \mathbb{E}\big(h(\mathbf{Z})\big) - \mathbb{E}\big(h(\mathbf{Z}')\big) \Big), \tag{1}$$

where $\mathcal{H}$ is a class of functions $h : \mathcal{X}_u \to \mathbb{R}$, which is specified as a reproducing kernel Hilbert Space (RKHS) associated with a continuous kernel function $K(\cdot, \cdot)$. From (1), when MMD is large, the distributions between $\mathbf{Z}$ and $\mathbf{Z}'$ appear dissimilar.

In NCD, the unlabeled dataset is predicted by taking the information from the conditional probability $\mathbb{P}_{Y_l|\mathbf{X}_l}$ (usually presented by a pretrained neural network) of a labeled dataset. For example, if the distributions of $\mathbb{P}_{Y_l|\mathbf{X}_l=\mathbf{X}_u}$ under $Y_u = c$ and $Y_u = c'$ are significantly different, then its overall distribution discrepancy is large, yielding that more information can be leveraged in NCD. On this ground, we use MMD to quantify the discrepancy of the labeled probability measure $\mathbb{P}_{Y_l|\mathbf{X}_l}$ on $\mathbf{X}_u$ under the unlabeled probability measure $\mathbb{Q}$, namely *transfer flow*.

**Definition 2 (Transfer Flow)** *The* transfer flow *of NCD prediction under $\mathbb{Q}$ based on the labeled conditional probability $\mathbb{P}_{Y_l|\mathbf{X}_l}$ is*

$$\textit{T-Flow}(\mathbb{Q}, \mathbb{P}) = \mathbb{E}_{\mathbb{Q}}\Big( \textit{MMD}_{\mathcal{H}}^2\big(\mathbb{Q}_{\mathbf{p}(\mathbf{X}_u)|Y_u}, \mathbb{Q}_{\mathbf{p}(\mathbf{X}_u')|Y_u'}\big) \Big), \tag{2}$$

*where $(\mathbf{X}_u, Y_u), (\mathbf{X}_u', Y_u') \sim \mathbb{Q}$ are independent copies, the expectation $\mathbb{E}_{\mathbb{Q}}$ is taken with respect to $Y_u$ and $Y_u'$ under $\mathbb{Q}$, and $\mathbf{p}(\mathbf{x})$ is the conditional probability under $\mathbb{P}_{Y_l|\mathbf{X}_l}$ on an unlabeled data $\mathbf{X}_u = \mathbf{x}$, defined as*

$$\mathbf{p}(\mathbf{x}) = \big( \mathbb{P}\big( Y_l = c \mid \mathbf{X}_l = \mathbf{x} \big) \big)_{c \in \mathcal{C}_l}^{\mathsf{T}}.$$

To summarize, *transfer flow* measures the overall discrepancy of $\mathbf{p}(\mathbf{X}_u)$ under different new classes of the unlabeled measure $\mathbb{Q}$, which indicates the informative flow from $\mathbb{P}$ to $\mathbb{Q}$. Note that the metric intrinsically quantifies the information based on data, and is independent with NCD methods. Lemma 1 shows the lower and upper bounds of *transfer flow*, and provides a theoretical justification of its effectiveness in measuring the similarity between labeled and unlabeled datasets.

**Lemma 1** *$\kappa := \max_{c \in \mathcal{C}_u} \mathbb{E}_{\mathbb{Q}}\big(\sqrt{K(\mathbf{p}(\mathbf{X}_u), \mathbf{p}(\mathbf{X}_u))}|Y_u = c\big) < \infty$, then $0 \leq \textit{T-Flow}(\mathbb{Q}, \mathbb{P}) \leq 4\kappa^2$. Moreover, $\textit{T-Flow}(\mathbb{Q}, \mathbb{P}) = 0$ if and only if $Y_u$ is independent with $\mathbf{p}(\mathbf{X}_u)$, that is, for any $c \in \mathcal{C}_u$:*

$$\mathbb{Q}\big(Y_u = c \mid \mathbf{p}(\mathbf{X}_u)\big) = \mathbb{Q}\big(Y_u = c\big), \tag{3}$$

*yielding that $\mathbf{p}(\mathbf{X}_u)$ is useless in NCD on $\mathbb{Q}$.*

Note that $\kappa$ can be explicitly computed for many common used kernels, for example, $\kappa = 1$ for a Gaussian or Laplacian kernel.

From Lemma 1, T-Flow$(\mathbb{Q}, \mathbb{P}) = 0$ is equivalent to $Y_u$ is independent with $\mathbf{p}(\mathbf{X}_u)$, which matches our intuition of no flow. Alternatively, if $Y_u$ is dependent with $\mathbf{p}(\mathbf{X}_u)$, we justifiably believe that the information of $Y_l|\mathbf{X}_l$ can be used to facilitate NCD, Lemma 1 tells that T-Flow$(\mathbb{Q}, \mathbb{P}) > 0$ in this case. Therefore, Lemma 1 reasonably suggests that the proposed *transfer flow* is an effective metric to detect if the supervised information in $\mathbb{P}$ is useful to NCD on $\mathbb{Q}$.

Next, we give a finite sample estimate of *transfer flow*. To proceed, we first rewrite *transfer flow* as follows.

$$\text{T-Flow}(\mathbb{Q}, \mathbb{P}) = \sum_{c,c' \in \mathcal{C}_u; c \neq c'} \left( \mathbb{Q}(Y_u = c, Y_u' = c') \text{MMD}_{\mathcal{H}}^2 \left( \mathbb{Q}_{\mathbf{p}(\mathbf{X}_u)|Y_u=c}, \mathbb{Q}_{\mathbf{p}(\mathbf{X}_u')|Y_u'=c'} \right) \right), \tag{4}$$

where the equality follows from the fact that $\text{MMD}_{\mathcal{H}}^2 \left( \mathbb{Q}_{\mathbf{p}(\mathbf{X}_u)|Y_u=c}, \mathbb{Q}_{\mathbf{p}(\mathbf{X}_u')|Y_u'=c} \right) = 0$.

Given an estimated probability $\widehat{\mathbb{P}}_{Y_l|\mathbf{X}_l}$ and an evaluation dataset $(\mathbf{x}_{u,i}, y_{u,i})_{i=1}^m$ under $\mathbb{Q}$, we assess $\mathbf{x}_{u,i}$ on $\widehat{\mathbb{P}}_{Y_l|\mathbf{X}_l}$ as $\widehat{\mathbf{p}}(\mathbf{x}_{u,i}) = \left( \widehat{\mathbb{P}}\left( Y_l = c|\mathbf{X}_l = \mathbf{x}_{u,i} \right) \right)_{c \in \mathcal{C}_u}^{\mathsf{T}}$, then the empirical *transfer flow* is computed as:

$$\widehat{\text{T-Flow}}(\mathbb{Q}, \mathbb{P}) = \sum_{c,c' \in \mathcal{C}_u; c \neq c'} \left( \frac{|\mathcal{I}_{u,c}||\mathcal{I}_{u,c'}|}{m(m-1)} \widehat{\text{MMD}}_{\mathcal{H}}^2 \left( \mathbb{Q}_{\widehat{\mathbf{p}}(\mathbf{X}_u)|Y_u=c}, \mathbb{Q}_{\widehat{\mathbf{p}}(\mathbf{X}_u')|Y_u'=c'} \right) \right), \tag{5}$$

where $\mathcal{I}_{u,c} = \{1 \leq i \leq m : y_{u,i} = c\}$ is the index set of unlabeled data with $y_{u,i} = c$, and $\widehat{\text{MMD}}_{\mathcal{H}}^2$ is defined as:

$$\widehat{\text{MMD}}_{\mathcal{H}}^2 (\mathbb{Q}_{\widehat{\mathbf{p}}(\mathbf{X}_u)|Y_u=c}, \mathbb{Q}_{\widehat{\mathbf{p}}(\mathbf{X}_u')|Y_u'=c'}) = \frac{1}{|\mathcal{I}_{u,c}|(|\mathcal{I}_{u,c}|-1)} \sum_{i,j \in \mathcal{I}_{u,c}; i \neq j} K\left( \widehat{\mathbf{p}}(\mathbf{x}_{u,i}), \widehat{\mathbf{p}}(\mathbf{x}_{u,j}) \right)$$
$$+ \frac{1}{|\mathcal{I}_{u,c'}|(|\mathcal{I}_{u,c'}|-1)} \sum_{i,j \in \mathcal{I}_{u,c'}; i \neq j} K\left( \widehat{\mathbf{p}}(\mathbf{x}_{u,i}), \widehat{\mathbf{p}}(\mathbf{x}_{u,j}) \right) - \frac{2}{|\mathcal{I}_{u,c}||\mathcal{I}_{u,c'}|} \sum_{i \in \mathcal{I}_{u,c}} \sum_{j \in \mathcal{I}_{u,c'}} K\left( \widehat{\mathbf{p}}(\mathbf{x}_{u,i}), \widehat{\mathbf{p}}(\mathbf{x}_{u,j}) \right).$$

**Remark 1** *Note that the definition of the proposed* transfer flow *in Definition 2 is based on the conditional probability* $\mathbf{p}(\mathbf{x}_u)$ *from* $\mathbb{P}$. *Yet, it can be extended to a more general representation* $\mathbf{s}(\mathbf{x}_u)$ *estimated based on supervised or self-supervised information from a labeled dataset. See the definition of* pseudo transfer flow *as follows.*

Furthermore, we define *pseudo transfer flow*, which is computed by a pretrained representation $\widehat{\mathbf{s}}(\mathbf{x})$ and a pseudo-label obtained from a clustering method applied to the representations (e.g., K-means, GMM, agglomerative, etc.). It is worth mentioning that *pseudo transfer flow* is more practical than *transfer flow* because that *transfer flow* requires the estimated probabilities and the true label $Y_u$, while *pseudo transfer flow* can apply to any pretrained representation and any pseudo label obtained from a clustering method.

**Definition 3 (Pseudo Transfer Flow)** *The* pseudo transfer flow *of NCD prediction under* $\mathbb{Q}$ *based on a pretrained representation* $\widehat{\mathbf{s}}(\mathbf{x})$ *is*

$$\widehat{Pseudo\text{-}T\text{-}Flow}(\mathbb{Q}, \mathbb{P}) = \sum_{c,c' \in \mathcal{C}_u; c \neq c'} \frac{|\widetilde{\mathcal{I}}_{u,c}||\widetilde{\mathcal{I}}_{u,c'}|}{m(m-1)} \widehat{MMD}_{\mathcal{H}}^2 \left( \mathbb{Q}_{\widehat{\mathbf{s}}(\mathbf{X}_u)|\widetilde{Y}_u=c}, \mathbb{Q}_{\widehat{\mathbf{s}}(\mathbf{X}_u')|\widetilde{Y}_u'=c'} \right) \tag{6}$$

*where* $\widetilde{\mathcal{I}}_{u,c} = \{1 \leq i \leq m : \widetilde{y}_{u,i} = c\}$ *is the index set of unlabeled data with* $\widetilde{y}_{u,i} = c$, $\widetilde{y}_{u,i}$ *is provided based on a clustering method on their representations* $\widehat{\mathbf{s}}(\mathbf{x}_{u,i})$, *and* $\widehat{\mathbf{s}}(\mathbf{x}_{u,i})$ *is the representation estimated from a supervised model or a self-supervised model.* $\widehat{MMD}_{\mathcal{H}}^2$ *is defined as in Section 3.2.*

## 4 Benchmark

To examine the validity of the proposed metric, we first construct a benchmark with various degrees of semantic similarity by using the hierarchical structure in ImageNet. We design two groups of numerical

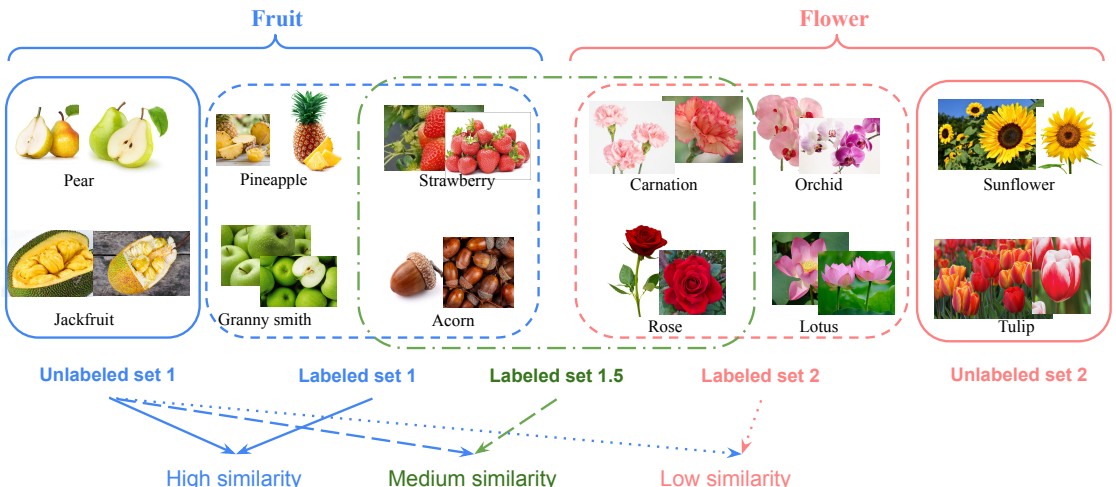

Figure 1: Illustration of how we construct the benchmark with varying levels of semantic similarity. Unlabeled set $U_1$ and labeled set $L_1$ are from the same superclass (fruit), whereas unlabeled set $U_2$ and labeled set $L_2$ belong to another superclass (flower). Labeled set $L_{1.5}$ is composed of half of $L_1$ and half of $L_2$. If both the labeled and unlabeled classes are derived from the same superclass, i.e., $(U_1, L_1)$ and $(U_2, L_2)$, we consider them a high semantic similarity split. In contrast, $(U_1, L_2)$ and $(U_2, L_1)$ are low semantic similarity splits, since the labeled and unlabeled classes are derived from distinct superclasses. In addition, we consider $(U_1, L_{1.5})$ and $(U_2, L_{1.5})$ to have medium semantic similarity because half of $L_{1.5}$ share the same superclass as $U_1$.

experiments: the consistency between NCD difficulty and degrees of semantic similarity (implemented by the hierarchical label structure) in Section 4.2.1, and the consistency between degrees of semantic similarity and the proposed (pseudo) *transfer flow* in Section 4.2.2.

## 4.1 Construction Principle

Unlike existing benchmarks, which only take into account the difficulty of NCD based on the labeled set in terms of the number of categories (e.g., Fini et al. (2021)) or the number of images in each category (e.g., Chi et al. (2022)), we argue that semantic similarity is also a significant factor influencing NCD performance.

Specifically, our proposed benchmark is based on the ENTITY-30 task (Santurkar et al., 2020), which contains 240 ImageNet classes in total, with 30 superclasses and 8 subclasses for each superclass. Three different semantic similarity levels (high, medium and low) are produced by leveraging ImageNet's underlying hierarchy. For example, as shown in Figure 1, despite the fact that the labeled (e.g., pineapple, strawberry) and unlabeled (e.g., pear, jackfruit) classes are disjoint, if they derive from the same superclass (i.e., fruit), they have a higher degree of semantic similarity. Conversely, when labeled (e.g., rose, lotus) and unlabeled (e.g., pear, jackfruit) classes are derived from distinct superclasses (i.e., labeled classes from flower while unlabeled classes from fruit), they are further apart semantically.

As a consequence, we define three labeled sets $L_1$, $L_{1.5}$, $L_2$ and two unlabeled sets $U_1$, $U_2$. The sets $L_1$ and $U_1$ are selected from the first 15 superclasses, where 6 subclasses of each superclass are assigned to $L_1$, and the other 2 are assigned to $U_1$. The sets $L_2$ and $U_2$ are created from the second 15 superclasses in a similar fashion. Finally, $L_{1.5}$ is created by taking classes half from $L_1$ and half from $L_2$. As a consequence, $(U_1, L_1)/(U_2, L_2)$ are closely related semantically (high), whereas $(U_1, L_2)/(U_2, L_1)$ are far apart (low), with $(U_1, L_{1.5})/(U_2, L_{1.5})$ in-between (medium). Additionally, we also provide four data settings on CIFAR100, two high-similarity settings and two low-similarity settings, by leveraging the hierarchical class structure of CIFAR100 similarly. Each case has 40 labeled classes and 10 unlabeled classes. A full list of the labeled and unlabeled sets can be found in Appendix E.

Table 1: Comparison of different combinations of labeled sets and unlabeled sets consisting of subsets of CIFAR100. The unlabeled sets are denoted $U_1$ and $U_2$, while the labeled sets are called $L_1$ and $L_2$. $U_1$ and $L_1$ share the same set of superclasses, similar for $U_2$ and $L_2$. Thus, the pairs $(U_1, L_1)$ and $(U_2, L_2)$ are close semantically, but $(U_1, L_2)$ and $(U_2, L_1)$ are far apart. We report the mean and standard deviation of the clustering accuracy across 10 runs for multiple NCD methods. The higher mean is bolded.

| Methods | Unlabeled set $U_1$ | | Unlabeled set $U_2$ | |
|---|---|---|---|---|
| | $L_1$ - high | $L_2$ - low | $L_1$ - low | $L_2$ - high |
| K-means (MacQueen et al., 1967) | **61.0 $\pm$ 1.1** | 37.7 $\pm$ 0.6 | 33.9 $\pm$ 0.5 | **55.4 $\pm$ 0.6** |
| DTC (Han et al., 2019) | **64.9 $\pm$ 0.3** | 62.1 $\pm$ 0.3 | 53.6 $\pm$ 0.3 | **66.5 $\pm$ 0.4** |
| RS (Han et al., 2020) | **78.3 $\pm$ 0.5** | 73.7 $\pm$ 1.4 | 74.9 $\pm$ 0.5 | **77.9 $\pm$ 2.8** |
| NCL (Zhong et al., 2021a) | **85.0 $\pm$ 0.6** | 83.0 $\pm$ 0.3 | 72.5 $\pm$ 1.6 | **85.6 $\pm$ 0.3** |
| UNO (Fini et al., 2021) | **92.5 $\pm$ 0.2** | 91.3 $\pm$ 0.8 | 90.5 $\pm$ 0.7 | **91.7 $\pm$ 2.2** |

Table 2: Comparison of different combinations of labeled sets and unlabeled sets on our proposed benchmark. Similar to the CIFAR-based experiments, $L_1$ is closely related to $U_1$ and $L_2$ is highly related to $U_2$. The third labeled set $L_{1.5}$ is constructed from half of $L_1$ and half of $L_2$, so in terms of similarity it is between $L_1$ and $L_2$. For all splits we report the mean and the standard deviation of the clustering accuracy across 10 runs for multiple NCD methods. The higher mean is bolded.

| Methods | Unlabeled set $U_1$ | | | Unlabeled set $U_2$ | | |
|---|---|---|---|---|---|---|
| | $L_1$ - high | $L_{1.5}$ - medium | $L_2$ - low | $L_1$ - low | $L_{1.5}$ - medium | $L_2$ - high |
| K-means | **41.1 $\pm$ 0.4** | 30.2 $\pm$ 0.4 | 23.3 $\pm$ 0.2 | 21.2 $\pm$ 0.2 | 29.8 $\pm$ 0.4 | **45.0 $\pm$ 0.4** |
| DTC | **43.3 $\pm$ 1.2** | 35.6 $\pm$ 1.3 | 32.2 $\pm$ 0.8 | 21.3 $\pm$ 1.2 | 15.3 $\pm$ 1.5 | **29.0 $\pm$ 0.8** |
| RS | **55.3 $\pm$ 0.4** | 50.3 $\pm$ 0.9 | 53.6 $\pm$ 0.6 | 48.1 $\pm$ 0.4 | 50.9 $\pm$ 0.6 | **55.8 $\pm$ 0.7** |
| NCL | **75.1 $\pm$ 0.8** | 74.3 $\pm$ 0.4 | 71.6 $\pm$ 0.4 | 61.3 $\pm$ 0.1 | 70.5 $\pm$ 0.8 | **75.1 $\pm$ 1.2** |
| UNO | **83.9 $\pm$ 0.6** | 81.0 $\pm$ 0.6 | 77.2 $\pm$ 0.8 | 77.5 $\pm$ 0.7 | 82.0 $\pm$ 1.7 | **88.4 $\pm$ 1.1** |

## 4.2 Experiments

### 4.2.1 Validating the Benchmark

**Experimental Settings** To suggest the effectiveness of the benchmark, we conduct experiments on 5 baselines, including K-means (MacQueen et al., 1967), DTC (Han et al., 2019), RS (Han et al., 2021), NCL (Zhong et al., 2021a) and UNO (Fini et al., 2021). We follow the baselines regarding hyperparameters and implementation details.

**Experimental Results** In Table 1 (CIFAR100), the gap between the high-similarity and the low-similarity settings is larger than 20% for K-means and reaches up to 12% for more advanced methods. Similarly, in Table 2 (ImageNet), the high-similarity settings generally obtain the best performance, followed by the medium and low settings. Under the unlabeled set $U_1$, $L_1$ achieves the highest accuracy, with around 2 - 17% improvement compared to $L_2$, and around 2 - 11% improvement compared to $L_{1.5}$. For the unlabeled set $U_2$, $L_2$ is the most similar set and obtains 8 - 14% improvement compared to $L_1$, and around 5 - 14% improvement compared to $L_{1.5}$.

**Conclusion** *Consistency between Semantic Similarity and Accuracy.* The above numerical results suggest a positive correlation between semantic similarity and NCD performance, which suggests that semantic similarity is a significant factor influencing NCD performance.

Table 3: Experiments on *transfer flow*. To obtain the standard deviation we recompute the *transfer flow* 10 times using bootstrap sampling. The results show that *transfer flow* is consistent with semantic similarity. The maximum *transfer flow* for the same unlabeled set is highlighted in bolded.

| Dataset | Unlabeled Set | Labeled Set | *Transfer flow* |
|---|---|---|---|
| CIFAR100 | $U_1$ | $L_1$ - high | **0.62 $\pm$ 0.01** |
| | | $L_2$ - low | 0.28 $\pm$ 0.01 |
| | $U_2$ | $L_1$ - low | 0.33 $\pm$ 0.01 |
| | | $L_2$ - high | **0.77 $\pm$ 0.02** |
| ImageNet | $U_1$ | $L_1$ - high | **0.71 $\pm$ 0.01** |
| | | $L_{1.5}$ - medium | 0.54 $\pm$ 0.01 |
| | | $L_2$ - low | 0.36 $\pm$ 0.01 |
| | $U_2$ | $L_1$ - low | 0.33 $\pm$ 0.00 |
| | | $L_{1.5}$ - medium | 0.50 $\pm$ 0.01 |
| | | $L_2$ - low | **0.72 $\pm$ 0.01** |

Table 4: Experiments on *pseudo transfer flow* under three clustering methods, i.e., K-means, GMM and agglomerative, each setting is repeated for 10 times. The maximum *pseudo transfer flow* is highlighted in bold for each baseline.

| Method | Unlabeled set $U_1$ | | | Unlabeled set $U_2$ | | |
|---|---|---|---|---|---|---|
| | $L_1$ - high | $L_{1.5}$ - medium | $L_2$ - low | $L_1$ - low | $L_{1.5}$ - medium | $L_2$ - high |
| K-means | **1.23 $\pm$ 0.03** | 1.02 $\pm$ 0.03 | 0.99 $\pm$ 0.02 | 0.96 $\pm$ 0.01 | 0.99 $\pm$ 0.03 | **1.24 $\pm$ 0.02** |
| GMM | **0.79 $\pm$ 0.01** | 0.69 $\pm$ 0.02 | 0.56 $\pm$ 0.02 | 0.58 $\pm$ 0.02 | 0.68 $\pm$ 0.04 | **0.91 $\pm$ 0.02** |
| Agglomerative | **1.17 $\pm$ 0.00** | 0.96 $\pm$ 0.00 | 0.87 $\pm$ 0.00 | 0.83 $\pm$ 0.00 | 0.89 $\pm$ 0.00 | **1.15 $\pm$ 0.00** |

### 4.2.2  Validating (Pseudo) *Transfer Flow*

**Experimental Settings**  We evaluate *transfer flow* and *pseudo transfer flow* on both CIFAR100 and our proposed ImageNet-based benchmark under different semantic similarity cases. We employ ResNet18 (He et al., 2016) as the backbone for both datasets following Han et al. (2019; 2021); Fini et al. (2021). Known-class data and unknown-class data are selected based on semantic similarity, as mentioned in Section 4. We first apply fully supervised learning to the labeled data for each dataset to obtain the pretrained model. Then, we feed the unlabeled data to the pretrained model to obtain its representation. Lastly, we calculate the *transfer flow/pseudo transfer flow* based on the pretrained model and the unlabeled samples' representation. Specifically, for *pseudo transfer flow*, we apply clustering methods to generate the pseudo labels. For the first step, batch size is set to 512 for both datasets. We use an SGD optimizer with momentum 0.9, and weight decay 1e-4. The learning rate is governed by a cosine annealing learning rate schedule with a base learning rate of 0.1, a linear warmup of 10 epochs, and a minimum learning rate of 0.001. We pretrain the backbone for 200/100 epochs for CIFAR-100/ImageNet.

**Experimental Results and Conclusions**  (i) *Consistency between (pseudo) transfer flow and semantic similarity.* Table 3 demonstrates the *transfer flow* under different data settings. In CIFAR100, settings with high semantic similarity tend to have higher *transfer flow* than those with low semantic similarity. Similarly, in ImageNet, settings with high semantic similarity have the highest *transfer flow*, followed by those with medium semantic similarity, while those with low semantic similarity have the lowest *transfer flow*. Table 4 shows the *pseudo transfer flow* under various baselines and datasets. This result supports the conclusion above that, for a given baseline, settings with higher semantic similarity tend to have higher *pseudo transfer flow*.

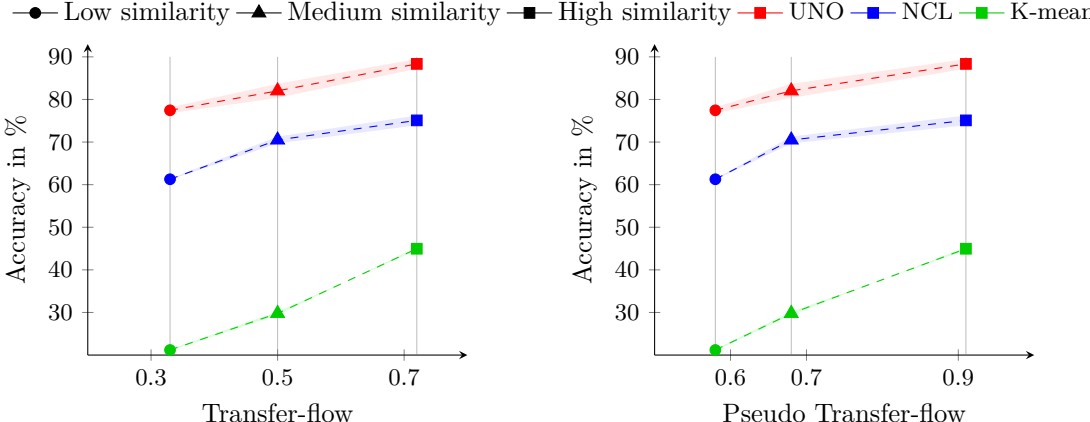

Figure 2: The experiments investigate the correlation between accuracy and *transfer flow/pseudo transfer flow*. We show three baselines, including UNO, NCL and K-means. On the left, we measure the *transfer flow* and the clustering accuracy. On the right, we replace *transfer flow* with *pseudo transfer flow* calculated from GMM clustering. As expected, there is a positive correlation between accuracy and *transfer flow* as well as *pseudo transfer flow*, which shows that *transfer flow* and *pseudo transfer flow* can demonstrate the difficulty of NCD tasks.

(ii) *Consistency between (pseudo) transfer flow and accuracy.* Figure 2 illustrates the relationship between semantic similarity, *transfer flow/pseudo transfer flow*, and NCD performance on ImageNet, where the same color corresponds to the same baseline. As expected, there is a consistent positive correlation between *transfer flow/pseudo transfer flow* and NCD accuracy, supporting the validity of *transfer flow/pseudo transfer flow* as a metric for quantifying semantic similarity and the difficulty of a particular NCD problem.

## 5 Supervised Knowledge May Hurt Performance

NCD is based on the idea that supervised knowledge from labeled data can be used to improve the clustering of unlabeled data. Yet, our empirical studies raise a possibility that supervised information from a labeled set may result in suboptimal outcomes compared to using exclusively self-supervised knowledge.

### 5.1 Empirical Experiments

We first conduct experiments in the following settings:

(a) $\mathbf{X}_u + \mathbf{X}_l$, using the images of unlabeled set and the labeled set but without labels.

(b) $\mathbf{X}_u + (\mathbf{X}_l, Y_l)$, using the unlabeled set and the whole labeled set, (i.e., standard NCD).

(c) $\mathbf{X}_u$, using the unlabeled set.

Particularly, for (a), even though we do not use the labels, we can still extract the knowledge of the labeled set via self-supervised learning. By comparing (a) and (b), we can analyze the performance gain from including supervised knowledge in the form of labels. For a better comparison, we also include part of experiments on (c) for estimating the total performance gain caused by adding the labeled set in Appendix (Table 9). In (c), NCD is degenerated to unsupervised learning (i.e., clustering on $\mathbf{X}_u$).

**Experimental Settings** We conduct experiments based on recent NCD methods, including RS (Han et al., 2021), NCL (Zhong et al., 2021a) and UNO (Fini et al., 2021) (the current state-of-the-art method in NCD). To perform settings (a), we make adjustments to the framework of each method with as minimal modifications as possible, enabling it to run fully self-supervised. For UNO, we replace ground truth labels $y_{l_{GT}}$ in the

labeled set with self-supervised pseudo labels $y_{l_{PL}}$, which are obtained by applying the Sinkhorn-Knopp algorithm (Cuturi, 2013). For NCL and RS, we replace the ground truth labels with labels obtained using K-means (MacQueen et al., 1967). More details on these modifications as well as the used hyperparameters can be found in Appendix B.2. We conduct experiments on our Imagenet-based benchmark, as describe in Section 4, where we define high, medium, and low similarity settings based on the hierarchical structure of the dataset and we evaluate these settings by *transfer flow*. Additionally, we conducted experiments on the CIFAR100 50-50, established by Fini et al. (2021), which randomly splits the CIFAR100 dataset (Krizhevsky, 2009) into 50 labeled and 50 unlabeled classes without considering semantic similarity.

**Experimental Results** As shown in Table 5, by comparing (a) and (b) under different baselines, we have the following numerical results:

- For RS, (a) outperforms (b) on all datasets, with 3% improvement in CIFAR100 and $\sim$10 - 15% improvement in ImageNet.

- For NCL, (b) exceeds (a) by $\sim$2.5% in CIFAR100, whereas in ImageNet, (a) surpasses (b) by $\sim$2 - 8% in all semantic similarity settings.

- For UNO, we discuss 3 settings,
    - In high - similarity settings, (i.e., $L_1 - U_1$), (a) performs $\sim$4% worse than (b).
    - In low - similarity settings, (i.e., $L_2 - U_1$ and $L_1 - U_2$), (a) outperforms (b) by $\sim$3% - 8%.
    - In medium - similarity settings, (i.e., $L_{1.5} - U_1$ and $L_{1.5} - U_2$), neither (a) nor (b) has an absolute advantage.

**Conclusion** *Supervision information with low semantic relevance may hurt NCD performance.* Based on the analysis of numerical results, we find that using self-supervised knowledge is significantly more advantageous than using supervised knowledge in both RS and NCL, while in UNO, supervised knowledge is beneficial when the labeled and unlabeled sets have a high degree of semantic similarity, but harmful when the semantic similarity is low.

Table 5: Comparison of using supervised knowledge or using exclusively self-supervised knowledge on CIFAR100 and our proposed benchmark. We present clustering mean and standard error on three recent methods, including RS, NCL and UNO (SOTA). Unexpectedly, $\mathbf{X}_u + \mathbf{X}_l$ outperforms $\mathbf{X}_u + (\mathbf{X}_l, Y_l)$ in both RS and NCL on ImageNet. For UNO, in CIFAR100-50 and low similarity case of our benchmark, $\mathbf{X}_u + \mathbf{X}_l$ can also get greater performance than $\mathbf{X}_u + (\mathbf{X}_l, Y_l)$. The empirical results indicate that supervised knowledge may have a negative impact on NCD performance. The higher mean is bolded.

| | Setting | CIFAR100-50 | Unlabeled set $U_1$ | | | Unlabeled set $U_2$ | | |
|---|---|---|---|---|---|---|---|---|
| | | | $L_1$ - high | $L_{1.5}$ - medium | $L_2$ - low | $L_1$ - low | $L_{1.5}$ - medium | $L_2$ - high |
| RS | $\mathbf{X}_u + \mathbf{X}_l$ | **42.8 ± 0.4** | **64.9 ± 0.2** | **64.5 ± 0.4** | **67.3 ± 0.9** | **62.9 ± 1.2** | **65.3 ± 0.5** | **67.4 ± 0.8** |
| | $\mathbf{X}_u + (\mathbf{X}_l, Y_l)$ | 39.2 ± 1.0 | 55.3 ± 0.4 | 50.3 ± 0.9 | 53.6 ± 0.6 | 48.1 ± 0.4 | 50.9 ± 0.6 | 55.8 ± 0.7 |
| NCL | $\mathbf{X}_u + \mathbf{X}_l$ | 50.9 ± 0.4 | **77.3 ± 0.4** | **75.2 ± 0.5** | **75.9 ± 0.6** | **77.3 ± 0.6** | **77.5 ± 0.7** | **83.2 ± 0.8** |
| | $\mathbf{X}_u + (\mathbf{X}_l, Y_l)$ | **53.4 ± 0.3** | 75.1 ± 0.8 | 74.3 ± 0.4 | 71.6 ± 0.4 | 61.3 ± 0.1 | 70.5 ± 0.8 | 75.1 ± 1.2 |
| UNO | $\mathbf{X}_u + \mathbf{X}_l$ | **64.1 ± 0.4** | 79.6 ± 1.1 | 79.7 ± 1.0 | **80.3 ± 0.3** | **85.3 ± 0.5** | **85.2 ± 1.0** | 89.2 ± 0.3 |
| | $\mathbf{X}_u + (\mathbf{X}_l, Y_l)$ | 62.2 ± 0.2 | **83.9 ± 0.6** | **81.0 ± 0.6** | 77.2 ± 0.8 | 77.5 ± 0.7 | 82.0 ± 1.7 | 88.4 ± 1.1 |

## 5.2 Practical Applications

Table 5 indicates that supervised knowledge from the labeled set may cause harm rather than gain, it is nature to ask whether to utilize supervised knowledge with labeled data or pure self-supervised knowledge without labels. Therefore, we provide two instructive solutions, including a practical metric (i.e., *pseudo transfer flow*) and a new method tuning the weighting between supervised and/or self-supervised knowledge.

Table 6: Results showing the link between *pseudo transfer flow* (PTF) and accuracy on novel classes (ACC). The *pseudo transfer flow* is computed based either on a supervised (SL) or self-supervised model (SSL), using ResNet18 in both cases. The accuracy is obtained using the standard NCD setting ($\mathbf{X}_u + (\mathbf{X}_l, Y_l)$) for supervised learning, and self-supervised NCD setting ($\mathbf{X}_u + \mathbf{X}_l$) for self-supervised model. The higher mean value is presented in bold, while the results within standard deviation of the average accuracy are not bolded.

|  | Model | High similarity | | Medium similarity | | Low similarity | |
|---|---|---|---|---|---|---|---|
|  |  | $L_1 - U_1$ | $L_2 - U_2$ | $L_{1.5} - U_1$ | $L_{1.5} - U_2$ | $L_2 - U_1$ | $L_1 - U_2$ |
| PTF | SSL | $0.96 \pm 0.01$ | $0.96 \pm 0.02$ | $\mathbf{1.14 \pm 0.02}$ | $\mathbf{1.19 \pm 0.01}$ | $\mathbf{1.05 \pm 0.03}$ | $\mathbf{1.25 \pm 0.03}$ |
|  | SL | $\mathbf{1.21 \pm 0.02}$ | $\mathbf{1.21 \pm 0.01}$ | $1.03 \pm 0.02$ | $0.98 \pm 0.03$ | $0.99 \pm 0.02$ | $0.96 \pm 0.01$ |
| ACC | SSL | $79.6 \pm 1.1$ | $89.2 \pm 0.3$ | $79.7 \pm 1.0$ | $\mathbf{85.2 \pm 1.0}$ | $\mathbf{80.3 \pm 0.3}$ | $\mathbf{85.3 \pm 0.5}$ |
|  | SL | $\mathbf{83.9 \pm 0.6}$ | $88.4 \pm 1.1$ | $81.0 \pm 0.6$ | $82.0 \pm 1.7$ | $77.2 \pm 0.8$ | $77.5 \pm 0.7$ |

### 5.2.1 Data Selection: Supervised or Self-supervised Knowledge?

We conduct the experiment to investigate the relationship between *pseudo transfer flow* and NCD's accuracy under various semantic similarity settings.

**Experimental Settings** We first perform supervised learning and self-supervised learning to achieve two pretrained models under varying data sets based on UNO (the best performing method in Section 5), respectively. Then, *pseudo transfer flow* is computed based on the two pretrained models.

**Experimental Results** In Table 6, PTF denotes *pseudo transfer flow* and ACC represents accuracy. We find that *pseudo transfer flow* is consistent with the accuracy under various datasets. For example, in $L_1$ - $U_1$, the *pseudo transfer flow* computed on the supervised model is larger than the one computed in the self-supervised model, which is consistent with the accuracy, where the supervised method outperforms the self-supervised one. Reversely, for $L_2$ - $U_1$, $L_1$ - $U_2$ and $L_{1.5}$ - $U_1$, the *pseudo transfer flow* computed on the self-supervised model is larger than the one computed in the supervised model, which is again consistent with their relative performance. $L_2$ - $U_2$ and $L_{1.5}$ - $U_1$ are omitted as their performance falls within the standard deviation of the average accuracy.

**Conclusion** *The proposed* pseudo transfer flow *can be used as a practical reference to infer what sort of data we want to use in NCD, image-only information,* $\mathbf{X}_u + \mathbf{X}_l$ *or image-label pairs,* $\mathbf{X}_u + (\mathbf{X}_l, Y_l)$ *of the labeled set.*

### 5.2.2 Data Combining: Weighting Supervised Knowledge

Rather than adopting a binary approach of either fully utilizing or disregarding supervised knowledge, it is essential to determine the optimal amount of supervised knowledge to incorporate.

**Method and Experimental Settings** Specifically, we first utilize self-supervised learning for pretraining rather than using supervised pretraning as UNO. Then, similar to UNO, we improve image representations by a contrastive learning framework (i.e., SwAV (Caron et al., 2020)). Different from UNO, we generate pseudo labels $y_{l_{PL}}$ for labeled data utilizing Sinkhorn-Knopp algorithm (Cuturi, 2013) and combine them with its corresponding ground truth labels $y_{l_{GT}}$. Then, the overall classification target of the labeled data is $y_l = \alpha y_{l_{GT}} + (1 - \alpha)y_{l_{PL}}$, where $\alpha \in [0, 1]$ represents the weight of the supervised component. Specifically, when $\alpha = 1$, our proposed method has the same target as UNO (Fini et al., 2021), but the pretraining is different.

**Experimental Results**

- From Figure 3, by comparing supervised (dotted lines) and self-supervised pretraining (dashed lines), we can see that in the high similarity setting, supervised pretraining is slightly better than self-

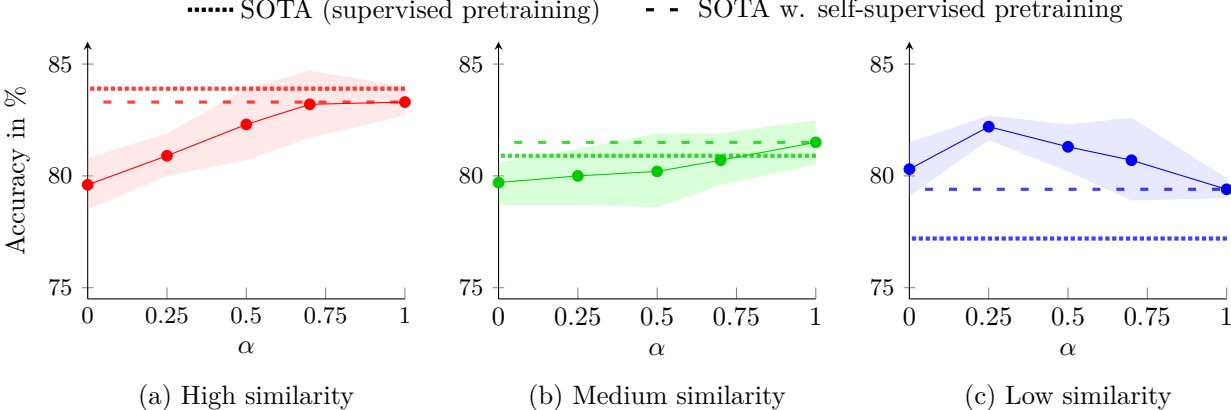

Figure 3: Experiments on combining supervised and self-supervise objectives, where $\alpha$ shows the weight of the supervised component. The experiments are carried out on our ImageNet-based benchmark with high, medium and low similarity settings, respectively. Dotted lines show the accuracy of the SOTA (UNO, using supervised pretraining) and dashed lines show the accuracy of SOTA when replacing the pretraining with self-supervised learning. In the low-similarity setting, a mix of supervised and self-supervised objectives outperforms either alone. Self-supervised pretraining outperforms supervised pretraining in low and medium similarity settings and is comparable in high similarity settings.

supervised pretraining. Insterestingly, in the medium similarity setting, self-supervised pretraining outperform supervised pretraining, and the advantage of self-supervised pretraining becomes more significant in the low similarity setting, with an improvement of approximately 2%.

- As observed in the low similarity setting, NCD accuracy demonstrates an increasing trend followed by a decreasing trend as the utilization of supervised knowledge ($\alpha$) rises, with an approximate enhancement of $2 - 3\%$ when $\alpha$ is set to 0.25 in comparison to the fully supervised ($\alpha = 1$) and exclusively self-supervised ($\alpha = 0$) training, and $\sim 5\%$ improvement compared to SOTA. Specifically, pure self-supervised training ($\alpha = 0$) surpasses fully supervised training ($\alpha = 1$) with $\sim 1\%$ improvement. In contrast, in the high-similarity setting, NCD accuracy demonstrates an upward trend with the increase in the level of supervised knowledge. However, in the medium similarity setting, the improvement is not substantial with an increase in supervised knowledge.

**Conclusion**

- *Supervised knowledge may lead to inferior performance during both pretraining and training.*

- *The effectiveness of incorporating supervised knowledge in the training process is closely related to the degree of similarity between the labeled and unlabeled datasets.* Therefore, in order to mitigate the occurrence of negative transfer, the use of supervised knowledge in NCD should be considered with regard to the appropriate amount rather than being employed in a naive manner.

## 6 Conclusion

We thoroughly investigate the effectiveness of supervised knowledge in the NCD task. We first introduce *transfer flow/pseudo transfer flow*, a metric for measuring the semantic similarity between two data sets. Then, we propose a comprehensive benchmark with varying levels of semantic similarity based on ImageNet for validating the proposed metric and verify that semantic similarity has a significant impact on NCD performance. By leveraging the metric and benchmark, we observe that supervised knowledge may lead to inferior performance in circumstances with low semantic similarity. Furthermore, we propose two practical applications: (i) *pseudo transfer flow* as a reference on what sort of data we aim to use. (ii) weighting supervised knowledge, which obtains ∼5% improvement under low similarity settings for ImageNet.

## 7 Acknowledgement

This work was supported in part by RGC-ECS 24302422, and CUHK Faulty of Science direct grant. The authors are grateful to reviewers and the Action Editor for their insightful comments and suggestions which have improved the manuscript significantly.

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

# A    Details of Section 3

## A.1    Technical Proofs

To proceed, we summarize all notations used in the paper in Table 7.

Table 7: Notation used in the paper.

| Notation | Description |
|---|---|
| $\mathbf{X}_l, \mathbf{X}_u$ | labeled data / unlabeled data |
| $y_l, y_u$ | label of labeled data / unlabeled data |
| $\mathcal{X}_l, \mathcal{X}_u$ | domain of labeled data / unlabeled data |
| $\mathcal{C}_l, \mathcal{C}_u$ | label set of labeled data / unlabeled data |
| $\mathbb{P}, \mathbb{Q}$ | probability measure of labeled data / unlabeled data |
| $\mathcal{L}_n = (\mathbf{X}_{l,i}, Y_{l,i})_{i=1,\cdots,n}$ | labeled dataset |
| $\mathcal{U}_m = (\mathbf{X}_{u,i})_{i=1,\cdots,m}$ | unlabeled dataset |
| $\mathcal{H}$ | reproducing kernel Hilbert space (RKHS) |
| $K(\cdot, \cdot)$ | kernel function |
| $(\mathbf{X}', Y')$ | independent copy of $(\mathbf{X}, Y)$ |
| $\widehat{\mathbb{P}}$ | estimated probability measure of labeled data |
| $\mathbb{E}_{\mathbb{Q}}$ | expectation with respect to the probability measure $\mathbb{Q}$ |
| $\mathbf{x}_{u,i}, y_{u,i}$ | the $i$-th unlabeled data |
| $\mathcal{I}_{u,c}$ | index set of unlabeled samples labeled as $y_{u,i} = c$ |

**Proof of Lemma 1.** We first show the upper bound of the transfer flow. According to Lemma 3 in Gretton et al. (2012), we have

$$
\begin{aligned}
\text{T-Flow}(\mathbb{Q}, \mathbb{P}) = \mathbb{E}_{\mathbb{Q}}\big(\text{MMD}^2(\mathbb{Q}_{\mathbf{p}(\mathbf{X}_u)|Y_u}, \mathbb{Q}_{\mathbf{p}(\mathbf{X}'_u)|Y'_u})\big) &= \mathbb{E}_{\mathbb{Q}}\Big(\big\|\mu_{\mathbb{Q}_{\mathbf{p}(\mathbf{X}_u)|Y_u}} - \mu_{\mathbb{Q}_{\mathbf{p}(\mathbf{X}'_u)|Y'_u}}\big\|_{\mathcal{H}}^2\Big) \\
&\leq \max_{c,c'\in\mathcal{C}_u}\big\|\mu_{\mathbb{Q}_{\mathbf{p}(\mathbf{X}_u)|Y_u=c}} - \mu_{\mathbb{Q}_{\mathbf{p}(\mathbf{X}'_u)|Y'_u=c'}}\big\|_{\mathcal{H}}^2 \leq 4\max_{c\in\mathcal{C}_u}\|\mu_{\mathbb{Q}_{\mathbf{p}(\mathbf{X}_u)|Y_u=c}}\|_{\mathcal{H}}^2 \\
&= 4\max_{c\in\mathcal{C}_u}\langle\mathbb{E}_{\mathbb{Q}}\big(K(\mathbf{p}(\mathbf{X}_u),\cdot)|Y_u=c\big), \mathbb{E}_{\mathbb{Q}}\big(K(\mathbf{p}(\mathbf{X}'_u),\cdot)|Y'_u=c\big)\rangle_{\mathcal{H}} \\
&= 4\max_{c\in\mathcal{C}_u}\mathbb{E}_{\mathbb{Q}}\big(\langle K(\mathbf{p}(\mathbf{X}_u),\cdot), K(\mathbf{p}(\mathbf{X}'_u),\cdot)\rangle_{\mathcal{H}}|Y_u=c, Y'_u=c\big) \\
&\leq 4\max_{c\in\mathcal{C}_u}\mathbb{E}_{\mathbb{Q}}\Big(\big\|K(\mathbf{p}(\mathbf{X}_u),\cdot)\big\|_{\mathcal{H}}\big\|K(\mathbf{p}(\mathbf{X}'_u),\cdot)\big\|_{\mathcal{H}}|Y_u=c, Y'_u=c\Big) \\
&= 4\max_{c\in\mathcal{C}_u}\mathbb{E}_{\mathbb{Q}}\big(\sqrt{K(\mathbf{p}(\mathbf{X}_u),\mathbf{p}(\mathbf{X}_u))}|Y_u=c\big)\mathbb{E}_{\mathbb{Q}}\big(\sqrt{K(\mathbf{p}(\mathbf{X}'_u),\mathbf{p}(\mathbf{X}'_u))}|Y'_u=c\big) \leq 4\kappa^2,
\end{aligned}
$$

where $\mu_{\mathbb{Q}_{\mathbf{p}(\mathbf{X}_u)|Y_u}} := \mathbb{E}_{\mathbb{Q}}\big(K(\mathbf{p}(\mathbf{X}_u),\cdot)|Y_u\big)$ is the kernel mean embedding of the measure $\mathbb{Q}_{\mathbf{p}(\mathbf{X}_u)|Y_u}$ (Gretton et al., 2012), the second inequality follows from the triangle inequality in the Hilbert space, the fourth equality follows from the fact that $\mathbb{E}_{\mathbb{Q}}$ is a linear operator, the second last inequality follows from the Cauchy-Schwarz inequality, and the last equality follows the reproducing property of $K(\cdot, \cdot)$.

Next, we show the if and only if condition for T-Flow$(\mathbb{Q}, \mathbb{P}) = 0$. Assume that $\mathbb{Q}(Y_u = c) > 0$ for all $c \in \mathcal{C}_u$. According to Theorem 5 in Gretton et al. (2012), we have

$$
\text{T-Flow}(\mathbb{Q}, \mathbb{P}) = 0 \quad \Longleftrightarrow \quad \mathbb{Q}\big(\mathbf{p}(\mathbf{x})|Y_u = c\big) = q_0(\mathbf{x}), \text{ for } c \in \mathcal{C}_u, \mathbf{x} \in \mathcal{X}_u.
$$

Note that

$$
1 = \sum_{c\in\mathcal{C}_u}\mathbb{Q}(Y_u = c|\mathbf{p}(\mathbf{x})) = \sum_{c\in\mathcal{C}_u}\frac{\mathbb{Q}(\mathbf{p}(\mathbf{x})|Y_u = c)\mathbb{Q}(Y_u = c)}{\mathbb{Q}(\mathbf{p}(\mathbf{x}))} = \sum_{c\in\mathcal{C}_u}\frac{q_0(\mathbf{x})\mathbb{Q}(Y_u = c)}{\mathbb{Q}(\mathbf{p}(\mathbf{x}))} = \frac{q_0(\mathbf{x})}{\mathbb{Q}(\mathbf{p}(\mathbf{x}))},
$$

yielding that $\mathbb{Q}\big(\mathbf{p}(\mathbf{x})|Y_u = c\big) = \mathbb{Q}(\mathbf{p}(\mathbf{x}))$, for $c \in \mathcal{C}_u, \mathbf{x} \in \mathcal{X}_u$. This is equivalent to,

$$\mathbb{Q}\big(Y_u = c|\mathbf{p}(\mathbf{x})\big) = \frac{\mathbb{Q}\big(\mathbf{p}(\mathbf{x})|Y_u = c\big)\mathbb{Q}(Y_u = c)}{\mathbb{Q}(\mathbf{p}(\mathbf{x}))} = \mathbb{Q}(Y_u = c).$$

This completes the proof. □

### A.2 Additional Experiments on the Robustness of Transfer Flow

To investigate the consistency of *transfer flow* across different kernels and bandwidths, we compare the *transfer flow* values of the Gaussian and Laplacian kernels with varying bandwidths ($h$). In this study, we consider Gaussian and Laplacian kernels, each with 5 different bandwidths $h$. We compute the bandwidth as:

$$h = \frac{\sum_{i=1}^{n} \sum_{j=1}^{n} \|\mathbf{x_i} - \mathbf{x_j}\|_2}{n(n-1)},$$

where $n$ is the number of samples and $x_i$ and $x_j$ represent the $i$-th and $j$-th samples, respectively. The physical meaning of bandwidth is the average distance between all possible pairs of data points in the whole dataset.

Our analysis reveals that the *transfer flow* values differ across the different kernels and bandwidths (as shown in Table 8). However, we consistently observe that the high similarity settings result in higher transfer leakage compared to the low similarity settings.

Table 8: Comparison of *transfer flow* values for Gaussian and Laplacian kernel functions and vary bandwidths (h) on CIFAR100. The high similarity settings consistently result in greater *transfer flow* compared to the low similarity settings. The highest value for each unlabeled set and kernel function is bolded.

| Kernel | Similarity | $\frac{1}{2^2}h$ | $\frac{1}{2}h$ | $h$ | $2h$ | $2^2h$ | Sum |
|---|---|---|---|---|---|---|---|
| Gaussian | High | **0.13 ± 0.00** | **0.15 ± 0.00** | **0.15 ± 0.00** | **0.11 ± 0.00** | **0.08 ± 0.00** | **0.62 ± 0.01** |
| | Low | 0.06 ± 0.00 | 0.07 ± 0.00 | 0.07 ± 0.00 | 0.05 ± 0.00 | 0.04 ± 0.00 | 0.28 ± 0.01 |
| Laplacian | High | **0.06 ± 0.00** | **0.11 ± 0.00** | **0.12 ± 0.00** | **0.09 ± 0.00** | **0.06 ± 0.00** | **0.43 ± 0.00** |
| | Low | 0.03 ± 0.00 | 0.05 ± 0.00 | 0.06 ± 0.00 | 0.05 ± 0.00 | 0.03 ± 0.00 | 0.21 ± 0.00 |

## B Details of Section 5

### B.1 Additional Experiments

Table 9: Comparison of different data settings on CIFAR100 and our proposed benchmark. We present clustering mean and standard error for each setting. (c) uses only the unlabeled set, whereas (a) uses both the unlabeled set and the labeled set's images without labels. (b) represents the standard NCD setting, i.e., using the unlabeled set and the whole labeled set. However, in CIFAR100-50 and low similarity case of our benchmark, (a) can get greater performance than (b). The higher mean is bolded.

| Setting | CIFAR100-50 | Unlabeled set $U_1$ | | | Unlabeled set $U_2$ | | |
|---|---|---|---|---|---|---|---|
| | | $L_1$ - high | $L_{1.5}$ - medium | $L_2$ - low | $L_1$ - low | $L_{1.5}$ - medium | $L_2$ - high |
| (c) $\mathbf{X}_u$ | 54.9 ± 0.4 | 70.5 ± 1.2 | 70.5 ± 1.2 | 70.5 ± 1.2 | 71.9 ± 0.3 | 71.9 ± 0.3 | 71.9 ± 0.3 |
| (a) $\mathbf{X}_u + \mathbf{X}_l$ | **64.1 ± 0.4** | 79.6 ± 1.1 | 79.7 ± 1.0 | **80.3 ± 0.3** | **85.3 ± 0.5** | **85.2 ± 1.0** | 89.2 ± 0.3 |
| (b) $\mathbf{X}_u + (\mathbf{X}_l, Y_l)$ | 62.2 ± 0.2 | **83.9 ± 0.6** | **81.0 ± 0.6** | 77.2 ± 0.8 | 77.5 ± 0.7 | 82.0 ± 1.7 | 88.4 ± 1.1 |

We also conduct full experiment on UNO with settings (a) - (c) to further examine the benefit of the self-supervised from the labeled set. In Table 9, by comparing (a) and (c), we find that NCD performance is consistently improved by incorporating more images (without labels) from a labeled set, with around 10% improvement in accuracy on CIFAR100. For our benchmark, setting (a) obtains an improvement

about 6% - 10% over (c) and the increase is more obvious in the low similarity settings. *This numerical results demonstrates that self-supervised knowledge from the labeled set is beneficial for NCD performance under varying semantic similarity.*

### B.2   Experimental Setup and Hyperparameters

In general, we follow the baselines regarding hyperparameters and implementation details unless stated otherwise. We repeat all experiments on CIFAR100 10 times and the ones on our proposed ImageNet-based benchmark 5 times, and report mean and standard deviation. All experiments are conducted using PyTorch and run on NVIDIA V100 GPUs.

**Adapting Baselines to the Self-supervised Setting**   Since experiment settings (1) and (2) do not make use of any labels, we need to adapt UNO (Fini et al., 2021) to work without labeled data. The standard UNO method conducts NCD in a two-step approach. In the first step, it applies a supervised pretraining on the labeled data only. The pretrained model is then used as an initialization for the second step, in which the model is trained jointly on both labeled and unlabeled data using one labeled head and multiple unlabeled heads. To achieve this, the logits of known and novel classes are concatenated and the model is trained using a single cross-entropy loss. Here, the targets for the unlabeled samples are taken from pseudo-labels, which are generated from the logits of the unlabeled head using the Sinkhorn-Knopp algorithm (Cuturi, 2013)

To adapt UNO to the fully unsupervised setting in (1), we need to remove all parts that utilize the labeled data. Therefore, in the first step, we replace the supervised pretraining by a self-supervised one, which is trained only on the unlabeled data. For the second step, we simply remove the labeled head, thus the method is degenerated to a clustering approach based solely on the pseudo-labels generated by the Sinkhorn-Knopp algorithm. For setting (2), we apply the self-supervised pretraining based on both unlabeled and labeled images to obtain the pretrained model in the first step. In the second step, we replace the ground-truth labels for the known classes with pseudo-labels generated by the Sinkhorn-Knopp algorithm based on the logits of these classes. Taken together, the updated setup utilizes the labeled images, but not their labels.

For NCL and RS, the modifications are similar to the ones done on UNO. Both methods consist of three steps, a self-supervised pretraining step, followed by another supervised pretraining and lastly the novel class discovery step. To adapt them to setting (2), we replace the first two steps with a single self-supervised pretraining step based on SwAV (Caron et al., 2020). For the last step, we simply replace the ground truth labels with labels obtained using k-means (MacQueen et al., 1967), while keeping the framework itself the same.

**Hyperparameters**   We conduct our experiments on CIFAR100 as well as our proposed ImageNet-based benchmark. All settings and hyperparameters are kept as close as possible as to the original baselines, including the choice of ResNet18 as the model architecture. We use SwAV as self-supervised pretraining for all experiments. The pretraining is done using the small batch size configuration of the method, which uses a batch size of 256 and a queue size of 3840. The training is run for 800 epochs, with the queue being enabled at 60 epochs for our ImageNet-based benchmark and 100 epochs for CIFAR100. To ensure a fair comparison with the standard NCD setting, the same data augmentations were used. In the second step of UNO, we train the methods for 500 epochs on CIFAR100 and 100 epochs for each setting on our benchmark. The experiments are replicated 10 times on CIFAR100 and 5 times on the developed benchmark, and the averaged performances and their corresponding standard errors are summarized in Table 5.

## C   Computation Cost

Overall, the methods we assessed have relatively low computational costs, primarily because of the utilization of a lightweight ResNet18 backbone across all methods. For instance, training a single split of our benchmark using ImageNet took approximately 8 hours for supervised pretraining and about 13 hours for the class discovery phase, all performed on a single Nvidia V100 GPU. During the inference phase, the methods manifest higher similarity since computation steps exclusive to the training process, such as the Sinkhorn-Knopp algorithm for UNO, or the computation of ranking statistics for RS, are excluded.

The complexity of (pseudo) *transfer flow* is $O((C_u)^2 * m^2)$, where $C_u$ denotes the number of classes in the unlabeled dataset and $m$ denotes the number of unlabeled samples.

## D   Discussion

The nature of the *dark knowledge* transferred from the labeled set is still mystery and we provide intuitive understandings on why supervised knowledge can have a negative impact on NCD performance.

**Bias / Conflicting information:** In cases where there is significant bias and conflicting information between the supervised knowledge $(Y_l|\mathbf{X_l})$ in the labeled data and the predictive information $(Y_u|\mathbf{X_u})$ in the unlabeled data, the utilization of supervised knowledge may lead to negative effects. Intuitively, supervised knowledge obtained from $\mathbf{X}_l, Y_l$ provides two pieces of information, including classification rule and improved representation, while self-supervised information from $\mathbf{X}_l$ primarily enhances representation. However, in scenarios with low semantic similarity or differing classification rules, the conflicting information present can pose challenges for the model to reconcile effectively.

**Limited generalization:** From the information bottleneck Saxe et al. (2019); Tishby et al. (2000); Shamir et al. (2010) perspective, feature space $\mathbf{X}$ has a larger dimension and contains richer information content. However, the incorporation of category information $Y$ may lead to the removal of information that is not related to category $Y$, which can result in a reduction in the dimension of the feature space. Furthermore, combined with the first point, when the bias is high, the removed feature space may overlap with the unlabelled features. This may be one of the reasons why incorporating self-supervised has shown performance improvement, while incorporating supervised information has led to a reduction.

# E   Detailed Benchmark Splits

Table 10: ImageNet class list of labeled split $L_1$ and unlabeled split $U_1$ of our proposed benchmark. As they share the same superclasses, they are highly related semantically. For each superclass, six classes are assigned to the labeled set and two to the unlabeled set. The labeled classes marked by the red box are also included in $L_{1.5}$, which shares half of its classes with $L_1$ and half with $L_2$.

| Superclass | Labeled Subclasses | Unlabeled Subclasses |
|---|---|---|
| garment | vestment, jean, academic gown, sarong, fur coat, apron | swimming trunks, miniskirt |
| tableware | wine bottle, goblet, mixing bowl, coffee mug, water bottle, water jug | plate, beer glass |
| insect | leafhopper, long-horned beetle, lacewing, dung beetle, sulphur butterfly, fly | admiral, grasshopper |
| vessel | wreck, liner, container ship, catamaran, trimaran, lifeboat | yawl, aircraft carrier |
| building | toyshop, grocery store, bookshop, palace, butcher shop, castle | beacon, mosque |
| headdress | cowboy hat, bathing cap, pickelhaube, bearskin, bonnet, hair slide | crash helmet, shower cap |
| kitchen utensil | cocktail shaker, frying pan, measuring cup, tray, spatula, cleaver | caldron, coffeepot |
| footwear | knee pad, sandal, clog, cowboy boot, running shoe, Loafer | Christmas stocking, maillot |
| neckwear | stole, necklace, feather boa, bow tie, Windsor tie, neck brace | bolo tie, bib |
| bony fish | puffer, sturgeon, coho, eel, rock beauty, tench | gar, lionfish |
| tool | screwdriver, fountain pen, quill, shovel, screw, combination lock | torch, padlock |
| vegetable | spaghetti squash, cauliflower, zucchini, acorn squash, artichoke, cucumber | cardoon, butternut squash |
| motor vehicle | beach wagon, trailer truck, limousine, police van, convertible, school bus | garbage truck, moped |
| sports equipment | balance beam, rugby ball, ski, horizontal bar, racket, dumbbell | tennis ball, croquet ball |
| carnivore | otterhound, flat-coated retriever, Italian greyhound, Shih-Tzu, basenji, black-footed ferret | Boston bull, Bedlington terrier |

Table 11: ImageNet class list of labeled split $L_2$ and unlabeled split $U_2$ of our proposed benchmark. As they share the same superclasses, they are highly related semantically. For each superclass, six classes are assigned to the labeled set and two to the unlabeled set. The labeled classes marked by the red box are also included in $L_{1.5}$, which shares half of its classes with $L_1$ and half with $L_2$.

| Superclass | Labeled Subclasses | Unlabeled Subclasses |
|---|---|---|
| fruit | corn, buckeye, strawberry, pear, Granny Smith, pineapple | acorn, jackfruit |
| saurian | African chameleon, Komodo dragon, alligator lizard, agama, green lizard, Gila monster | banded gecko, American chameleon |
| barrier | stone wall, chainlink fence, breakwater, dam, bannister, picket fence | worm fence, turnstile |
| electronic equipment | cassette player, modem, printer, monitor, computer keyboard, pay-phone | dial telephone, microphone |
| serpentes | green snake, boa constrictor, green mamba, ringneck snake, thunder snake, king snake | rock python, garter snake |
| dish | hot pot, burrito, potpie, meat loaf, cheeseburger, mashed potato | hotdog, pizza |
| home appliance | espresso maker, toaster, washer, space heater, vacuum, microwave | dishwasher, Crock Pot |
| measuring instrument | wall clock, barometer, digital watch, hourglass, magnetic compass, analog clock | digital clock, parking meter |
| primate | indri, siamang, baboon, capuchin, chimpanzee, howler monkey | patas, Madagascar cat |
| crustacean | rock crab, king crab, crayfish, American lobster, Dungeness crab, spiny lobster | fiddler crab, hermit crab |
| musical instrument | organ, acoustic guitar, French horn, electric guitar, upright, maraca | violin, grand piano |
| arachnid | black and gold garden spider, wolf spider, harvestman, tick, black widow, barn spider | tarantula, scorpion |
| aquatic bird | dowitcher, goose, albatross, limpkin, white stork, red-backed sandpiper | drake, crane |
| ungulate | hippopotamus, hog, llama, hartebeest, ox, gazelle | warthog, zebra |
| passerine | house finch, magpie, goldfinch, indigo bunting, chickadee, brambling | bulbul, water ouzel |

Table 12: Labeled Split of CIFAR100 used in Section 4. We construct data settings based on its hierarchical class structure. $U_1$-$L_1$/$U_2$-$L_2$ are share the same superclasses.

| Superclass | Labeled Subclasses ($L_1$) | Unlabeled Subclasses ($U_1$) |
|---|---|---|
| aquatic_mammals | dolphin, otter, seal, whale | beaver |
| fish | flatfish, ray, shark, trout | aquarium_fish |
| flower | poppy, rose, sunflower, tulip | orchids |
| food containers | bowl, can, cup, plate | bottles |
| fruit and vegetables | mushroom, orange, pear, sweet_pepper | apples |
| household electrical devices | keyboard, lamp, telephone, television | clock |
| household furniture | chair, couch, table, wardrobe | bed |
| insects | beetle, butterfly, caterpillar, cockroach | bee |
| large carnivores | leopard, lion, tiger, wolf | bear |
| large man-made outdoor things | castle, house, road, skyscraper | bridge |
| Superclass | Labeled Subclasses ($L_2$) | Unlabeled Subclasses ($U_2$) |
| large natural outdoor scenes | forest, mountain, plain, sea | cloud |
| large omnivores and herbivores | cattle, chimpanzee, elephant, kangaroo | camel |
| medium-sized mammals | porcupine, possum, raccoon, skunk | fox |
| non-insect invertebrates | lobster, snail, spider, worm | crab |
| people | boy, girl, man, woman | baby |
| reptiles | dinosaur, lizard, snake, turtle | crocodile |
| small mammals | mouse, rabbit, shrew, squirrel | hamster |
| trees | oak_tree, palm_tree, pine_tree, willow_tree | maple |
| vehicles 1 | bus, motorcycle, pickup_truck, train | bicycle |
| vehicles 2 | rocket, streetcar, tank, tractor | lawn-mower |

