# OpenReview forum: "Supervised Knowledge May Hurt Novel Class Discovery Performance"
_TMLR — Accepted by TMLR_

### Review · Reviewer_wTCF · 2023-02-10

**Summary Of Contributions:**

Novel Class Discovery (NCD) aims to infer novel categories in an unlabeled dataset by leveraging prior knowledge from a labeled set of related but disjoint classes. This paper examines the question of whether supervised knowledge is always helpful for NCD at different levels of semantic relevance. The inadequacy of existing NCD literature is revealed, which usually assumes that supervised knowledge is beneficial. A novel metric, transfer leakage, is proposed to measure the semantic similarity between labeled and unlabeled datasets. A new benchmark based on ImageNet class hierarchies is also built to show the validity of the proposed metric. A pseudo-version of the transfer leakage is developed as a practical reference to decide if supervised knowledge should be used in NCD. The effectiveness of the proposed modules is verified by extensive empirical studies.

**Audience:**

Yes

**Claims And Evidence:**

No

**Requested Changes:**

Thanks to the authors for the work. I hope the authors answer the questions mentioned in the above weaknesses. Although I like the idea of the paper, I’m unsure about the method clarity and especially the experimental correctness.

**Strengths And Weaknesses:**

For strengths,
-	The paper identified and studied a meaningful question: Is supervised knowledge always helpful at different levels of semantic relevance? The studied problem is well-motivated. The effect of different levels of semantic relevance is rarely studied before.
-	The proposed metrics transfer leakage/pseudo transfer leakage are interesting and can be applied to the analysis of other problems as well.
-	The conclusion of the paper. supervised knowledge can harm the performance of NCD and the combination of supervised and self-supervised knowledge works better, will contribute to the community, and may open up new research directions.
-	The experiments are conducted in various setups. The experimental design is comprehensive and clear.

For weaknesses,
-	The article has some corresponding analysis for transfer leakage. For the more practical pseudo-transfer leakage, can there be some analysis and guarantee as well?
-	The description of transfer leakage is not very clear. Especially it took me a while to understand the confusing notations such as p(x), p(X_u), X_u=x, X_l=x. Would like to see this section written in more detail.
-	For the experiments, some results seem inconsistent. For example, the performance of UNO under L1.5-U1 is 81.0±0.5 in table 2, but the performance becomes 80.9±1.2 in table 5. Same for L2-U2.
-	Some findings seem problematic. For table 6, the higher pseudo-transfer leakage doesn’t always imply better performance (L2-U2 and L1.5-U1). Why does the paper simply say “pseudo transfer leakage is consistent with the accuracy under various datasets”?
-	For figure 3, why setting alpha to zero or one can already outperform SOTA accuracy? In fact, no matter what the value of alpha is, the performance is always better than SOTA. The figure requires further justifications.

---

> ### Author Response · Authors · 2023-02-17
> **Response to Reviewer wTCF**
>
> >**Q1:** The article has some corresponding analysis for transfer leakage. For the more practical pseudo-transfer leakage, can there be some analysis and guarantee as well?
>
> **A1:** Thanks for the comment. The consistency study of pseudo-transfer leakage is now empirically examined in Table 4. Specifically, experiments on pseudo transfer leakage under three clustering methods, i.e., K-means, GMM and agglomerative indicate the consistency between pseudo transfer leakage and semantic similarity.
>
> >**Q2:** The description of transfer leakage is not very clear. Especially it took me a while to understand the confusing notations such as p(x), p(X_u), X_u=x, X_l=x. Would like to see this section written in more detail.
>
> **A2:** Thank you for the comment. We have now carefully proofread the related section and polished some mathematical statements.
>
> >**Q3:** For the experiments, some results seem inconsistent. For example, the performance of UNO under L1.5-U1 is 81.0±0.5 in table 2, but the performance becomes 80.9±1.2 in table 5. Same for L2-U2.
>
> **A3:** Thank you for pointing this out. This is indeed a mistake that occurred during entering the data into the tables. We checked the numbers again to make sure such mistakes do not appear in the updated version.
>
> >**Q4:** Some findings seem problematic. For table 6, the higher pseudo-transfer leakage doesn’t always imply better performance (L2-U2 and L1.5-U1). Why does the paper simply say “pseudo transfer leakage is consistent with the accuracy under various datasets”?
>
> **A4:** For these two cases, the results are within standard deviation of the average accuracy. Thus, we can not make a statistical conclusion.  We also added more discussion to clarification.
>
> >**Q5:** For figure 3, why setting alpha to zero or one can already outperform SOTA accuracy? In fact, no matter what the value of alpha is, the performance is always better than SOTA. The figure requires further justifications.
>
> **A5:** Thanks for pointing that out. The confusion is caused by different usage of pretraining for SOTA (supervised pretraining) and the proposed method (self-supervised pretraining).  Following your suggestion, we have now added more experiments to include the SOTA with self-supervised pretraining (dashed lines) in Figure 3 to clarification. The major conclusions are itemized as follows based on the updated Figure 3.
> * First, as indicated in the updated Figure 3, by comparing supervised (dotted lines) and self-supervised pretraining (dashed lines), as you mentioned, we can see that self-supervised pretraining outperforms supervised pretraining with $\sim$2\% improvement in the low similarity setting. However, there is no significant difference in medium and high similarity settings. This suggests that similar conclusions continue to hold in NCD with pretraining, that is, using supervised knowledge in pretraining may also potentially harm NCD performance.
> * Second, as reported in the original papers, RS and NCL use self-supervised pretraining, while UNO uses supervised pretraining. And we found that UNO gets improvement under self-supervised pre-training.
> * Finally, we would like to indicate that the updated experiments echo the conclusion that the effectiveness of incorporating supervised knowledge in the training process depends on the degree of similarity between the labeled and unlabeled datasets.
>     * Low-similarity setting, NCD accuracy demonstrates an increasing trend followed by a decreasing trend as the utilization of supervised knowledge ($\alpha$) rises, with an approximate enhancement of $2-3$\% when $\alpha$ is set to $0.25$ in comparison to the fully supervised ($\alpha=1$) and exclusively self-supervised ($\alpha=0$) training and $\sim5$\% improvement compared to SOTA.
>     * Medium similarity setting, the improvement is not substantial with increased supervised knowledge.
>      * High-similarity setting, NCD accuracy demonstrates an upward trend with the increase in the level of supervised knowledge.

---

### Review · Reviewer_LiEt · 2023-02-12

**Summary Of Contributions:**

This paper considers the NCD problem and thinks about it more deeply. In Chi et. al. (2022), although they show the solvability of NCD, quantitive analysis is not explored. This paper fills this gap and contributes to the NCD field a lot.

**Audience:**

Yes

**Broader Impact Concerns:**

No concern regarding the broader impact.

**Claims And Evidence:**

Yes

**Requested Changes:**

See above.

**Strengths And Weaknesses:**

Pros:

This paper considers a significant problem, NCD, and provides a more detailed analysis of NCD.

A new metric is proposed to show the similarity of labeled and unlabeled data.

Extensive experiments are provided to validate the performance of the proposed method.

A new benchmark is proposed for NCD.

Cons:

As suggested by recent literature on MMD, kernel selection is important for it. How does the kernel selection influence your methods? Like changing the bandwidth of the Gaussian kernel.

In Chi et. al. (2022), they have an analysis regarding the solvability of NCD. Can the authors provide more comparison between your work and Chi et. al.?

---

> ### Author Response · Authors · 2023-02-17
> **Response to Reviewer LiEt**
>
> >**Q1:** As suggested by recent literature on MMD, kernel selection is important for it. How does the kernel selection influence your methods? Like changing the bandwidth of the Gaussian kernel.
>
> **A1:** Thanks for the insightful comment. A universal kernel, including Gaussian and Laplace kernels, is required to ensure the consistency of MMD. In our paper, we show the sum of 5 Gaussian kernels with different bandwidth (h), and each bandwidth is shown below. Following your suggestion, we added more experiments to examine the consistency over different kernels and bandwidths on CIFAR100.
>
> $h=\frac{\sum\_{i=1}^{n}\sum\_{j=1}^{n}\left\lVert \mathbf{x\_i - x\_j)} \right\rVert\_2}{n(n-1)}$, where $n$ denotes the number of samples, $x_i$ and $x_j$ represent the $i$-th sample and $j$-th sample, respectively. The physical meaning of bandwidth is the average of $L_2$ distance between all possible pairs of data points in the whole dataset.
>
> While the values of the proposed transfer leakage vary across different kernels and bandwidths (see Table 1), **the high similarity case consistently exhibits larger transfer leakage than the low similarity case.**
>
> ---
>
> **Table 1:  Transfer leakage vary across different kernels and bandwidths**
> |                   |                 | $\frac{1}{2^2}h$ | $\frac{1}{2}h$ |   $h$  | $2*h$ | $2^2*h$ |      Sum    |
> |-------------------|-----------------|:------------------------:|:----------------------:|:--------------:|:---------------:|:-----------------:|:------------:|
> |  Gaussian kernel  | High similarity |      $0.13 \pm 0.001$      |     $0.15 \pm 0.001$     | $0.15 \pm 0.001$ |  $0.11 \pm 0.001$ |   $0.08 \pm 0.001$  |  $0.62\pm0.01$ |
> |                   |  Low similarity |      $0.06 \pm 0.000$      |     $0.07 \pm 0.000$     | $0.07 \pm 0.000$ |  $0.05 \pm 0.000$ |   $0.04 \pm 0.000$  | $0.28\pm 0.01$ |
> | Laplacian kernel  | High similarity |      $0.06 \pm 0.000$      |     $0.11 \pm 0.001$     | $0.12 \pm 0.001$ |  $0.09 \pm 0.001$ |   $0.06 \pm 0.000$  |  $0.43\pm0.00$ |
> |                   |  Low similarity |     $ 0.03 \pm 0.000 $    |     $0.05 \pm 0.000$     | $0.06 \pm 0.000$ |  $0.05 \pm 0.000$ |   $0.03 \pm 0.000$  |  $0.21\pm0.00$ |
>
>
>
>
> ---
>
> >**Q2:** In Chi et. al. (2022), they have an analysis regarding the solvability of NCD. Can the authors provide more comparison between your work and Chi et. al.?
>
> **A2:** Thank you for the comment and bringing the reference for our attention. We have included them in Section 2, along with the comments on the distinctions.
> * Chi et. al. (2022) defined the solvability of NCD as if the optimal theoretical performance on the unlabeled dataset can be obtained by using the information from the labelled dataset. Especially, Condition (D) in Chi et. al. (2022) assumes that optimal transformation of labeled data overlaps the counterpart of unlabeled data, thus the best performance can be obtained simultaneously for both datasets. However, the theoretical conditions are quite difficult to verify in practice.
> * In contrast, our goal is not to justify if the theoretically optimal performance is achievable or not, yet positively investigate the effect from the labled dataset on NCD under different levels of semantic similarities. More specifically, to address the question if the labled dataset can improve the NCD performance, and provide the potential solutions. We have added more discussion in Section 2.

---

### Review · Reviewer_xz1z · 2023-02-12

**Summary Of Contributions:**

The paper mainly proposes a new viewpoint for novel class discovery problem: supervised knowledge may hurt novel class discovery performance. To verify the viewpoint, the paper first proposes a novel metric, called transfer leakage, to measure the semantic similarity between datasets. Then, the paper conducts experiments to demonstrate that supervised knowledge weakens novel class discovery performance when the semantic similarity between labeled and unlabeled datasets is low. Finally, the paper proposes two solutions to improve existing novel class discovery methods.

**Audience:**

Yes

**Claims And Evidence:**

No

**Requested Changes:**

Questions/suggestions:
- The paper said that “supervised knowledge may hurt novel class discovery performance”, and are there any potential reason why supervised knowledge hurts novel class discovery performance?
- In the problem definition, the paper sets up that samples in labeled and unlabeled datasets do not have same labels. However, in related works such as UNO, samples in labeled and unlabeled datasets share some labels. Why different papers choose different settings on labels in labeled and unlabeled datasets?
- In the definition 1 in Section 3.1, it is said “novel class discovery aims to predict the labels of the unlabeled dataset”. It seems that we only need to find the new classes in the unlabeled dataset, but do not need to classify the samples from the definition. However, in experiments, accuracy is used as evaluation metric, which shows that the category judgment of each sample is needed. Is the description in the definition proper?
- In the paper, the computation costs of different methods and transfer leakage/pseudo transfer leakage are not discussed.
- In section 5.1, there are 3 settings. But why there are the experiment results of (c) only appear in appendix?
- The experiment results in Section 5.2.1 mention error margins, is there any quantitative definition of error margin in the paper?
- Some figures/tables are not put at the right place. For example, Figure 1 is in page 3, but it is first mentioned in page 6.
- In the last sentence in conclusion, “a straightforward method” should be briefly introduced.
- Why the semantic similarity quantification is named as “transfer leakage”? The “leakage” shows some information that should not be spread is transferred.


**Strengths And Weaknesses:**

Strength:
- The viewpoint that supervised knowledge may hurt novel class discovery performance is pioneering and instructive for novel class discovery tasks.
- The proposed transfer leakage/pseudo transfer leakage can reflect the semantic similarity between labeled and unlabeled datasets, which can provide guidance for the usage of supervised data in novel class discovery problems.
- The approach the paper proposed obtains good results and outperforms the state-of-the-art method UNO.

Weaknesses:
- Although the proposed transfer leakage can provide guidance for handling novel class discovery problems, I cannot find an essential role of transfer leakage in the experiment. I mean that if we delete the contents about transfer leakage in Section 5, we can still obtain the conclusion that supervised knowledge may hurt novel class discovery performance from the experiments and provide solution in Section 5.2.2.
- From the experimental results in Section 5.2.2, we can see that from ~3%-~5% improvement in the experiment, ~2% improvement should be owed to self-supervised pretraining. Why is self-supervised pretraining not introduced in detail in the paper? Are there any existing works that use self-supervised pretraining to improve novel class discovery performance?

---

> ### Author Response · Authors · 2023-02-17
> **Response to Reviewer xz1z （1/3）**
>
> >**W1:** Although the proposed transfer leakage can provide guidance for handling novel class discovery problems, I cannot find an essential role of transfer leakage in the experiment. I mean that if we delete the contents about transfer leakage in Section 5, (i) we can still obtain the conclusion that supervised knowledge may hurt novel class discovery performance from the experiments and (ii) provide solution in Section 5.2.2.
>
> **A1:** Thanks for the comment. We would like to point out that the full version of the conclusion from our experiments is: supervised knowledge (with low similarity) may hurt novel class discovery performance. Toward this end, the proposed transfer leakage is involved in both experiments (Section 5.1) and the potential solutions (Section 5.2).
> - In Table 5, the high, medium, and low similarity settings are evaluated and defined by transfer leakage. We thus conclude that supervision information with low semantic relevance may hurt NCD performance. This is because “low similarity” is a subjective and qualitative notion, and it is more reasonable to measure similarity via transfer leakage.
> - In addition, transfer leakage is independent of the employed NCD method and solely depends on the data distribution. The validity of transfer leakage as a similarity measurement is supported by the benchmark proposed in Section 4.
> - Based on the empirical results in Table 6, transfer leakage implies a solution in Section 5.2.1, we can quickly decide on using supervised or self-supervised knowledge from the labeled set by calculating transfer leakage under different pre-training models rather than intensive tuning on the supervised weight $\alpha$.
>
> >**W2:** From the experimental results in Section 5.2.2, we can see that from 3%-5% improvement in the experiment, ~2% improvement should be owed to self-supervised pretraining. Why is self-supervised pretraining not introduced in detail in the paper? Are there any existing works that use self-supervised pretraining to improve novel class discovery performance?
>
> **A2:** Thanks for the suggestion. Following your suggestion, we have now added more experiments to include the SOTA with self-supervised pretraining (dashed lines) in Figure 3 for clarification. The major conclusions are itemized as follows based on the updated Figure 3.
> * First, as indicated in the updated Figure 3, by comparing supervised (dotted lines) and self-supervised pretraining (dashed lines), as you mentioned, we can see that self-supervised pretraining outperforms supervised pretraining with $\sim$2\% improvement in the low similarity setting. However, there is no significant difference in medium and high similarity settings. This suggests that similar conclusions continue to hold in NCD with pretraining, that is, **using supervised knowledge in pretraining may also potentially harm NCD performance.**
> * Second, as reported in the original papers, RS and NCL use self-supervised pretraining, while UNO uses supervised pretraining. And we found that UNO gets improvement under self-supervised pre-training.
> * Finally, we would like to indicate that the updated experiments echo the conclusion that the effectiveness of incorporating supervised knowledge in the training process depends on the degree of similarity between the labeled and unlabeled datasets.
>     * Low-similarity setting, NCD accuracy demonstrates an increasing trend followed by a decreasing trend as the utilization of supervised knowledge ($\alpha$) rises, with an approximate enhancement of $2-3$\% when $\alpha$ is set to $0.25$ in comparison to the fully supervised ($\alpha=1$) and exclusively self-supervised ($\alpha=0$) training and $\sim5$\% improvement compared to SOTA.
>    * Medium similarity setting, the improvement is not substantial with increased supervised knowledge.
>    * High-similarity setting, NCD accuracy demonstrates an upward trend with the increase in the level of supervised knowledge.

---

> > ### Author Response · Authors · 2023-02-17
> > **Response to Reviewer xz1z (2/3)**
> >
> > >**Q1:** The paper said that “supervised knowledge may hurt novel class discovery performance”, and are there any potential reason why supervised knowledge hurts novel class discovery performance?
> >
> > **A1:** Thank you for the comment. There could be several potential reasons why incorporating supervised knowledge may negatively impact novel class discovery performance. Here, we share our intuitive understandings and also update the Discussion in the Appendix.
> >
> > * **Bias / conflicting information**: In cases where there is significant bias and conflicting information between the supervised knowledge ($Y_l | X_l$) in the labeled data and the predictive information ($Y_u | X_u$) in the unlabeled data, the utilization of supervised knowledge may lead to negative effects. Intuitively, supervised knowledge obtained from $({X}_l, Y_l)$ provides two pieces of information, including classification rule and improved representation, while self-supervised information from ${X}_l$ primarily enhances representation. However, in scenarios with low semantic similarity or differing classification rules, the conflicting information present can pose challenges for the model to reconcile effectively.
> >
> > * **Limited generalization**: From the information bottleneck [1,2,3] perspective, feature space $X$ has a larger dimension and contains richer information content. However, the incorporation of category information $Y$ may lead to the removal of information that is not related to category $Y$, which can result in a reduction in the dimension of the feature space. Furthermore, combined with the first point, when the bias is high, the removed feature space may overlap with the unlabelled features. This may be one of the reasons why incorporating self-supervised has shown performance improvement, while incorporating supervised information has led to a reduction.
> >
> > [1] On the information bottleneck theory of deep learning, NIPS, 2017
> > [2] The information Bottleneck Method, 1999;
> > [3] Learning and generalization with information bottleneck, 2000
> >
> > >**Q2**: In the problem definition, the paper sets up that samples in labeled and unlabeled datasets do not have same labels. However, in related works such as UNO, samples in labeled and unlabeled datasets share some labels. Why different papers choose different settings on labels in labeled and unlabeled datasets?
> >
> > **A2:** Thank for the comment. We actually follow the same settings as other novel class discovery (NCD) papers, including UNO[1], RS[2], NCL[3], DTC[4] et al. In the NCD setting, labeled/known classes and unlabeled/unknown classes are disjoint. In contrast, the Generalized Category Discovery (GCD) approach relaxes this assumption, permitting unlabeled data to share the same known classes as the labeled set. GCD[5] was first proposed in 2022, and related works that employ this approach include OpenCon [6] and ORCA [7]. DTC initially introduced the NCD setting in 2019. We leave pursuing this topic in GCD as future work.
> >
> > [1]  A unified objective for novel class discovery. In ICCV, 2021.
> > [2]  Autonovel: Automatically discovering and learning novel visual categories.  In TPAMI, 2021.
> > [3] Neighborhood contrastive learning for novel class discovery. In CVPR, 2021.
> > [4] Learning to discover novel visual categories via deep transfer clustering. In CVPR, 2019.
> > [5] Generalized Category Discovery. In CVPR, 2022.
> > [6] Opencon: open-world contrastive learning. In TMLR, 2022.
> > [7] Open-world Semi-supervised Learning, In ICLR, 2022.
> >
> > >**Q3:** In the definition 1 in Section 3.1, it is said “novel class discovery aims to predict the labels of the unlabeled dataset”. It seems that we only need to find the new classes in the unlabeled dataset, but do not need to classify the samples from the definition. However, in experiments, accuracy is used as evaluation metric, which shows that the category judgment of each sample is needed. Is the description in the definition proper?
> >
> > **A3:** Sorry for the confusion. The NCD indeed aims to classify the samples in the unlabeled dataset. Following your suggestion, we have now revised the related statement in Definition 1 as “novel class discovery aims to predict the label $Y_u$ of each unlabeled instance $X_u$ given $\mathcal{L}_n$ and $\mathcal{U}_m$.”

---

> > > ### Author Response · Authors · 2023-02-17
> > > **Response to Reviewer xz1z (3/3)**
> > >
> > > >**Q4:** In the paper, the computation costs of different methods and transfer leakage/pseudo transfer leakage are not discussed.
> > >
> > > **A4:** Thanks for the comment and we added the computation costs in the updated revision.
> > > * Overall, the methods we assessed have relatively low computational costs, primarily because of the utilization of a lightweight ResNet18 backbone across all methods. For instance, training UNO on a single split of our benchmark using ImageNet took approximately 8 hours for supervised pretraining and about 13 hours for the class discovery phase, all performed on a single Nvidia V100 GPU.
> > > * During the inference phase, the methods manifest higher similarity since computation steps exclusive to the training process, such as the Sinkhorn-Knopp algorithm for UNO, or the computation of ranking statistics for RS, are excluded.
> > > * The complexity of (pseudo)transfer leakage is $O((C_u)^2 * m^2)$, where $C_u$ denotes the number of classes in the unlabeled dataset and $m$ denotes the number of unlabeled samples.
> > >
> > > >**Q5:** In section 5.1, there are 3 settings. But why there are the experiment results of (c) only appear in appendix?
> > >
> > > **A5:** We decided to only put the results of (c) in the appendix because this setting is not related to the question of whether or not labeled information/supervised knowledge supports novel class discovery.
> > > (c) uses only the unlabeled set, and the main objective is to compare the performance improvement resulting from incorporating labeled datasets, X_l or (X_l, Y_l).
> > >
> > > >**Q6:** The experiment results in Section 5.2.1 mention error margins, is there any quantitative definition of error margin in the paper?
> > >
> > > **A6:** Sorry for the confusion. Error margin used in the experiments is the standard deviation of the average accuracy. To clarify, we have revised “error margin” as “standard deviation of average accuracy” in places.
> > >
> > > >**Q7:** Some figures/tables are not put at the right place. For example, Figure 1 is in page 3, but it is first mentioned in page 6.
> > >
> > > **A7:** Thank you for pointing this out. We tried to improve the layout such that figures and tables are displayed closer to the referencing text in the updated version.
> > >
> > > >**Q8:** In the last sentence in conclusion, “a straightforward method” should be briefly introduced.
> > >
> > > **A8:** Agreed, we updated this in the revision.
> > >
> > > >**Q9:** Why the semantic similarity quantification is named as “transfer leakage”? The “leakage” shows some information that should not be spread is transferred.
> > >
> > > **A9:** Following your suggestion, we will revise “transfer leakage” as “transfer flow” after the discussion and will update it in the revised version.

---

### Decision · Action_Editors · 2023-04-20

**Recommendation:** Accept with minor revision

**Comment:**

See comments above.

**Audience:**

This paper would interest TMLR audience working on novel class discovery, and related problems in semi-supervised learning. It should interest practitioners trying to extract value from unlabeled data and to be able to analyze when this may or may not work based on measures of discrepancy introduced in this paper.

**Claims And Evidence:**

Novel class discovery (NCD) refers to the setting where a small labeled dataset is used to infer new related classes in an unlabeled dataset. This is closely reminiscent of semi-supervised learning where unlabeled data only helps under the so-called manifold, cluster or low-density separation assumptions (See Semisupervised learning book, Chapelle 2006).  The paper questions NCD literature that typically assumes positive transfer from the labeled set, and proposes an MMD based discrepancy measure semantic similarity between labeled and unlabeled datasets that is predictive of positive transfer. In particular, when this measure is low, the supervised knowledge may actually end up hurting. The reviewers requested clarity on the technical definitions and statistical significance in the reported experiments, how parameters of the MMD measure influence the results, computation costs of different methods etc. These were clarified during the review process, with more experimental evidence added. Hence, the premise and claims of the paper are reasonably well supported.